# Captured by Captions: On Memorization and its Mitigation in CLIP Models

**Wenhao Wang**[1]**, Adam Dziedzic**[1]**, Grace C. Kim**[2]**, Michael Backes**[1]**, Franziska Boenisch**[1*]
[1]CISPA, [2]Georgia Institute of Technology

## Abstract

Multi-modal models, such as CLIP, have demonstrated strong performance in aligning visual and textual representations, excelling in tasks like image retrieval and zero-shot classification. Despite this success, the mechanisms by which these models utilize training data, particularly the role of memorization, remain unclear. In uni-modal models, both supervised and self-supervised, memorization has been shown to be essential for generalization. However, it is not well understood how these findings would apply to CLIP, which incorporates elements from both supervised learning via captions that provide a supervisory signal similar to labels, and from self-supervised learning via the contrastive objective. To bridge this gap in understanding, we propose a formal definition of memorization in CLIP (CLIPMem) and use it to quantify memorization in CLIP models. Our results indicate that CLIP's memorization behavior falls between the supervised and self-supervised paradigms, with "mis-captioned" samples exhibiting highest levels of memorization. Additionally, we find that the text encoder contributes more to memorization than the image encoder, suggesting that mitigation strategies should focus on the text domain. Building on these insights, we propose multiple strategies to reduce memorization while at the same time improving utility—something that had not been shown before for traditional learning paradigms where reducing memorization typically results in utility decrease.

## 1 Introduction

Multi-modal models, such as CLIP (Radford et al., 2021), have demonstrated strong performance in representation learning. By aligning visual and textual representations, these models achieve state-of-the-art results in tasks like image retrieval (Baldrati et al., 2022a;b), visual question answering (Pan et al., 2023; Song et al., 2022), and zero-shot classification (Radford et al., 2021; Ali & Khan, 2023; Wang et al., 2023; Zhang et al., 2022). Despite these successes, the mechanisms by which multi-modal models leverage their training data to achieve good generalization remain underexplored.

In uni-modal setups, both supervised (Feldman, 2020; Feldman & Zhang, 2020) and self-supervised (Wang et al., 2024b), machine learning models have shown that their ability to *memorize* their training data is essential for generalization. It was indicated that, in supervised learning, memorization typically occurs for mislabeled samples, outliers (Bartlett et al., 2020; Feldman, 2020; Feldman & Zhang, 2020), or data points that were seen towards the end of training (Jagielski et al., 2022), while in self-supervised learning, high memorization is experienced particularly for atypical data points (Wang et al., 2024b). However, it is unclear how these findings extend to models like CLIP which entail elements from both supervised learning (through captions as supervisory signals) and self-supervised learning (through contrastive loss functions).

Existing definitions of memorization offer limited applicability to CLIP and therefore cannot fully address the gap in understanding. The standard definition from supervised learning (Feldman, 2020) relies on one-dimensional labels and the model's ability to produce confidence scores for these labels, whereas CLIP outputs high-dimensional representations. While the SSLMem metric (Wang et al., 2024b), developed for self-supervised vision models, could, in principle, be applied to CLIP's vision encoder outputs, it neglects the text modality, which is a critical component of CLIP. Additionally,

---
*Correspondence to boenisch@cispa.de

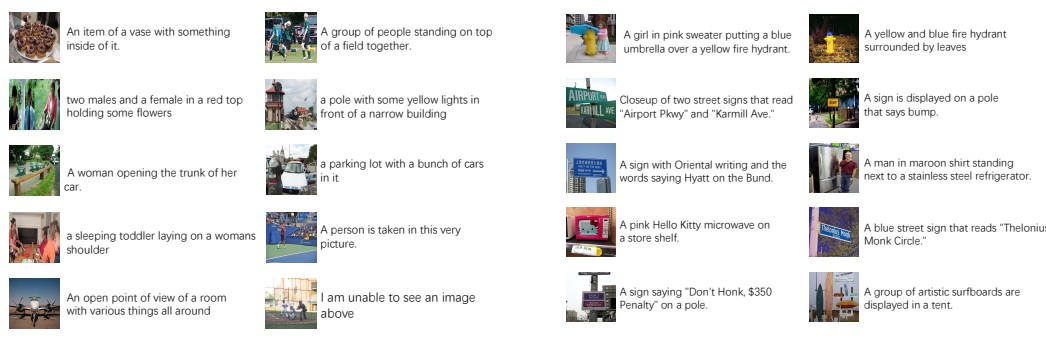

(a) Most Memorized: CLIPMem > 0.89       (b) Least Memorized: CLIPMem ≈ 0.0

Figure 1: **Examples of data with different levels of memorization.** Higher memorization scores indicate stronger memorization. We observe that atypical or distorted images, as well as those with incorrect or imprecise captions, experience higher memorization compared to standard samples and easy-to-label images with accurate captions. Results are obtained on OpenCLIP (Ilharco et al., 2021), with encoders based on the ViT-Base architecture trained on the COCO dataset.

measuring memorization in only one modality, or treating the modalities separately, risks diluting the signal and under-reporting memorization. Our experimental results, as shown in Section 4.3, confirm this concern. Therefore, new definitions of memorization tailored to CLIP's multi-modal nature are necessary.

The only existing empirical work on quantifying memorization in CLIP models (Jayaraman et al., 2024) focuses on Déjà Vu memorization (Meehan et al., 2023), a specific type of memorization. The success of their method relies on the accuracy of the integrated object detection method and on the availability of an additional public dataset from the same distribution as CLIP's training data, limiting practical applicability. To overcome this limitation, we propose *CLIPMem* that measures memorization directly on CLIP's output representations. Specifically, it compares the alignment—*i.e.,* the similarity between representations—of a given image-text pair in a CLIP model trained with the pair, to the alignment in a CLIP model trained on the same data but without the pair.

In our empirical study of memorization in CLIP using CLIPMem, we uncover several key findings. First, examples with incorrect or imprecise captions ("mis-captioned" examples) exhibit the highest levels of memorization, followed by atypical examples, as illustrated in Figure 1. Second, removing these samples from training yields significant improvements in CLIP's generalization abilities. These findings are particularly noteworthy, given that state-of-the-art CLIP models are usually trained on large, uncurated datasets sourced from the internet with no guarantees regarding the correctness of the text-image pairs. Our results highlight that this practice not only exposes imprecise or incorrect data pairs to more memorization, often recognized as a cause for increased privacy leakage (Carlini et al., 2019; 2021; 2022; Song et al., 2017; Liu et al., 2021), but that it also negatively affects model performance. Furthermore, by disentangling CLIP's two modalities, we are able to dissect how memorization manifests within each. Surprisingly, we find that memorization does not affect both modalities alike, with memorization occurring more in the text modality than in the vision modality. Building on these insights, we propose several strategies to reduce memorization while simultaneously improving generalization—a result that has not been observed in traditional supervised or self-supervised learning, where any reduction of memorization causes decreases in performance. Finally, at a deeper level, our analysis of the model internals, following Wang et al. (2024a), shows that CLIP's memorization behavior sits between that of supervised and self-supervised learning. Specifically, neurons in early layers are responsible for groups of data points (*e.g.,* classes), similar to models trained using supervised learning, while neurons in later layers memorize individual data points, as seen in self-supervised learning.

In summary, we make the following contributions:

• We propose CLIPMem, a metric to measure memorization in multi-modal vision language models.

- Through extensive evaluation, we identify that "mis-captioned" and "atypical" data points experience the highest memorization, and that the text encoder is more responsible for memorization than the image encoder.

- Based on our insights, we propose and evaluate multiple strategies to mitigate memorization in CLIP. We show that in CLIP, contrary to traditional supervised and self-supervised learning, a reduction of memorization does not need to imply a decrease in performance.

## 2 BACKGROUND AND RELATED WORK

**CLIP**. Contrastive Language-Image Pretraining (CLIP) (Radford et al., 2021) trains multi-modal encoders to map text-image pairs into a shared latent space with semantically equal representations. The core of CLIP is a two-encoder architecture with an image encoder $f_{\text{img}}$ and a text encoder $f_{\text{txt}}$ that are trained to maximize the similarity between the image and text features for correct text-image pairs, while minimizing the similarity for incorrect pairs. This is achieved using a contrastive loss function $\mathcal{L}$ defined as:

$$\mathcal{L} = -\frac{1}{N}\sum_{i=1}^{N}\log\frac{\exp(\text{sim}(f_{\text{img}}(x_i), f_{\text{txt}}(y_i))/\tau)}{\sum_{j=1}^{N}\exp(\text{sim}(f_{\text{img}}(x_i), f_{\text{txt}}(y_j))/\tau)},$$

where $\text{sim}(\cdot, \cdot)$ is the cosine similarity, $\tau$ is the temperature parameter, and $N$ is the batch size. This training makes CLIP versatile across various downstream tasks, including image classification, retrieval, captioning, and object recognition. There are different versions of CLIP. The popular Language augmented CLIP (LaCLIP) (Fan et al., 2023) augments original CLIP by introducing text augmentations during training in addition to the image augmentations (crops) performed in the original CLIP training to reduce overfitting. We study the impact of this practice on memorization and find it to be a suitable mitigation method.

**Memorization.** Memorization refers to a model's tendency to store specific details of individual training examples, rather than generalizing patterns across the dataset (Zhang et al., 2016; Arpit et al., 2017; Chatterjee, 2018; Feldman, 2020). This becomes problematic when models memorize sensitive data, as it has been shown to increase privacy risks (Carlini et al., 2019; 2021; 2022; Song et al., 2017). To date, memorization has been studied within *single modalities* for supervised and self-supervised learning. In **supervised learning**, it has been shown that models tend to memorize *mislabeled* (Feldman, 2020), *difficult*, or *atypical* examples (Arpit et al., 2017; Sadrtdinov et al., 2021), and that this memorization improves generalization, especially on long-tailed data (Feldman, 2020; Feldman & Zhang, 2020). Similar findings have been observed in **self-supervised learning** (SSL) in the vision domain (Wang et al., 2024b), where atypical samples experience high memorization, and a reduction of memorization in SSL encoders leads to decreased performance in various downstream tasks, such as classification, depth-estimation, and segmentation. A connection between memorization and generalization has also been observed in the language domain (Antoniades et al., 2024; Tirumala et al., 2022). In contrast to our work, these papers consider single-modality models. How those insights transfer to multi-modal models remains unclear.

**Memorization in self-supervised learning.** Our CLIPMem builds on concepts from the SSLMem metric introduced by Wang et al. (2024b). This metric measures the memorization of an individual data point $x$ by an SSL encoder, based on the alignment of representations from augmented views of $x$. Let $f : \mathbb{R}^n \to \mathbb{R}^d$ be an SSL encoder trained using an SSL algorithm $\mathcal{A}$ on an unlabeled dataset $S = \{x_i\}_{i=1}^m$. The data augmentations are represented as $\text{Aug}(x) = \{a(x)|a \in \text{Aug}\}$, where $a$ is a transformation function applied to the data point $x$, mapping from $\mathbb{R}^n \to \mathbb{R}^n$. The encoder's output representation for a given data point $x$ is denoted as $f(x)$. For a trained SSL encoder $f$, the alignment loss for a data point $x$ is defined as

$$\mathcal{L}_{\text{align}}(f, x) = \mathbb{E}_{x', x'' \sim \text{Aug}(x)}[d(f(x'), f(x''))], \tag{1}$$

where $x', x''$ are augmented views of $x$ and $d(\cdot, \cdot)$ is a distance metric, typically the $\ell_2$ distance. SSLMem is then defined as

$$\text{SSLMem}(x) = \mathbb{E}_{g \sim \mathcal{A}(S \setminus x)}\mathcal{L}_{\text{align}}(g, x) - \mathbb{E}_{f \sim \mathcal{A}(S)}\mathcal{L}_{\text{align}}(f, x) \tag{2}$$

with $f$ being an SSL encoder trained with data point $x$, and $g$, an encoder trained without $x$ but otherwise on the same dataset. While this framework measures memorization using alignment loss

for single-modality encoders, this approach is unsuitable to leverage the signal over both modalities from multi-modal encoders like CLIP, as we also highlight empirically in Section 4.3. However, we can build on the main concepts from SSLMem to define a new metric that can evaluate memorization in CLIP, by considering both image and text representations, as we will detail in Section 3.

**Memorization in CLIP**. Even though CLIP is a widely used vision-language encoder, there has been limited work on measuring memorization in CLIP. The only existing work (Jayaraman et al., 2024) applies the empirical Déjà Vu memorization framework from (Meehan et al., 2023) to CLIP. It measures memorization by computing the overlap between unique objects in potentially memorized images and their nearest neighbors—identified in the CLIP embedding space—from a public dataset. However, the reliance on external public data from the same distribution, along with the required accuracy of the object detection (which may not perform well for all samples, especially atypical ones (Kumar et al., 2023; Dhamija et al., 2020), limits the applicability of this approach. We further expand on this in Appendix A.1. In contrast, our CLIPMem operates directly on CLIP's output representations and returns a joint score over both modalities.

## 3 Defining Memorization over Multi-Modal Encoders

### 3.1 Problem Setup

Consider a single image-text pair $(I, T)$ from a dataset $S$ and two CLIP models: a model $f$ and a reference model $g$, trained on dataset $S$ and $S' = S \setminus \{(I, T)\}$, respectively. We aim to quantify the memorization of $(I, T)$ in $f$, trained on this data point, by leveraging model $g$ not trained on the data point but otherwise on the same data, in a *leave-one-out* style of defining memorization (Feldman, 2020). We denote the image encoder in CLIP as $f_{\text{img}} : \text{Image} \rightarrow \mathbb{R}^d$ and the text encoder as $f_{\text{txt}} : \text{Text} \rightarrow \mathbb{R}^d$. For the image-text pair $(I, T)$, we denote with $f_{\text{img}}(I)$ the output representation of $f$'s image encoder on image $I$ and with $f_{\text{txt}}(T)$ the output representation of $f$'s text encoder on text $T$. To evaluate the *alignment* between the image and text representations, *i.e.,* to quantify how similar the two representations are, we use cosine similarity $sim(f_{\text{img}}(I), f_{\text{txt}}(T))$, as defined in the original CLIP paper (Radford et al., 2021).

### 3.2 Alignment with Contrastive Objective

During training, the contrastive objective in CLIP maximizes the cosine similarity for correct image-text pairs while minimizing the cosine similarity for all the other $N - 1$ incorrect pairs in any given training mini-batch with $N$ training samples. This means that for a given image $I$ and text $T$, the training objective pulls $f_{\text{img}}(I)$ and $f_{\text{txt}}(T)$ closer together in the latent space, while pushing $f_{\text{img}}(I)$ away from the representations of all other $N - 1$ unrelated texts, and $f_{\text{txt}}(T)$ away from all other images. Hence, the intuition is that the quality of alignment in $f$, unlike in uni-modal self-supervised learning (Wang et al., 2024b), depends not only on the model's ability to create well-aligned text and image representations for a given text-image pair, but also on its ability to create distant representations for the $N - 1$ other representations.

To formalize this intuition into a metric that quantifies the alignment of $f$ on the image-text pair $(I, T)$, we define $\widehat{T_{test}}$ as a set of $N - 1$ randomly chosen testing samples that were not used in training $f$ or $g$. Furthermore, when applicable, we denote random augmentations of the training data—*e.g.,* text augmentations in versions like LaCLIP (Fan et al., 2023)—as $T' \sim \text{Aug}(T)$ for texts and $I' \sim \text{Aug}(I)$ for images. Then, we define the alignment score of $f$ on $(I, T)$ as

$$
\begin{aligned}
\mathcal{A}_{\text{align}}(f, I, T) = \underset{(I',T')\sim\text{Aug}(I,T)}{\mathbb{E}} & \left[ \text{sim}(f_{\text{img}}(I'), f_{\text{txt}}(T')) \right] \\
- \underset{(\_,t)\in\widehat{T_{test}}}{\mathbb{E}} & \left[ \text{sim}(f_{\text{img}}(I), f_{\text{txt}}(t)) \right] - \underset{(i,\_)\in\widehat{T_{test}}}{\mathbb{E}} \left[ \text{sim}(f_{\text{img}}(i), f_{\text{txt}}(T)) \right],
\end{aligned}
\tag{3}
$$

where high scores indicate a better alignment of $f$ on $(I, T)$. In case no text augmentations are applied, as in standard CLIP training, the first term is calculated only over $T$.

### 3.3 Defining Memorization in CLIP

Given our definition of alignment scores, we can define our CLIPMem in a similar vein to the definition of memorization in supervised learning (Feldman, 2020), in the leave-one-out style. Given the image-text pair $(I, T)$ from dataset $S$ and two CLIP models, $f$ and $g$, trained on dataset $S$ and $S' = S \setminus \{(I, T)\}$, respectively, we define CLIPMem as

$$\text{CLIPMem}(I, T) = \mathcal{A}_{\text{align}}(f, I, T) - \mathcal{A}_{\text{align}}(g, I, T). \tag{4}$$

If a model $f$ has a significantly higher alignment score than model $g$ on $(I, T)$, this means that $f$ memorizes this data point. Note that taking the difference between $f$ and $g$ is crucial to get a solid estimate of memorization. This is because without "context", a high or low alignment score of $f$ does not express much information. The alignment of $f$ can be high without memorizing $(I, T)$, for example, if $(I, T)$ is a simple (but not memorized) training example. In this case, the reference model $g$ will also have a high score, such that the difference is again small. Thanks to this design of our CLIPMem, it will then correctly report low memorization.

## 4 Empirical Evaluation

### 4.1 Experimental Setup

**Models and training.** We build our experiments on OpenCLIP (Cherti et al., 2023), an open-source Python version of Open-CLIP (Ilharco et al., 2021). The standard architecture used for the experiments builds on ViT-Base, but we also include experiments using ViT-Large. We train the model on the COCO dataset (Lin et al., 2014). Since COCO is much smaller than OpenCLIP's standard training datasets, we reduce the training batch size to 128 and increase the epoch number from 32 to 100 to achieve similar performance. All other settings strictly follow OpenCLIP. For training DINO, as an example of an SSL vision encoder, we follow the default setting of Caron et al. (2021). The supervised model is trained as a multi-label classifier, also based on ViT-Base (with an additional fully connection layer) based on the first-level annotation captions in the COCO dataset. A full specification of our experimental setup is detailed in Appendix A.2. Additional experiments for measuring memorization on the BLIP (Li et al., 2022) model are presented in Appendix A.6.

**Datasets.** We use COCO (Lin et al., 2014), CC3M (Sharma et al., 2018), and the YFCC100M (Thomee et al., 2016a) datasets to pre-train the OpenCLIP models. For the CC3M dataset, we randomly sample 75000 examples from the total of 2.91M data points. We evaluate the models by testing the linear probing accuracy on ImageNet (Deng et al., 2009) with an added classification layer trained on top of the output representations. We use the YFCC100M dataset to simulate an infinite data regime, *i.e.,* using a single training run where no data point is repeated whereas we train iteratively using CC3M and COCO.

**Measuring memorization.** We follow Wang et al. (2024b) to approximate our CLIPMem. Since training a separate pair of models for every data point whose memorization we aim to measure would be computationally intractable, we measure memorization of multiple data points at the same time. Therefore, we divide the original training set in four subsets: (1) $S_S$, data points that both model $f$ and $g$ were trained on, (2) $S_C$, data points used only for training $f$, (3) $S_I$, data points used only for training $g$, and (4) $S_E$, external "test" data points that none of the models was trained on. Note that $|S_C| = |S_I|$, such that $f$ and $g$ have the same number of training data points in total. For our experiments, following a similar approach to Wang et al. (2024b), we want to strike a balance when choosing the size of $S_C$. If the size is too large, then $f$ and $g$ might differ too much and not yield a strong memorization signal, but if it is too small, we would only have a memorization signal for too few data points. Concretely, for COCO and CC3M, we set $|S_S| = 65000$ and $|S_C| = |S_I| = |S_E| = 5000$. Memorization is reported as an average over all data points in $S_C$ for model $f$, or per individual data point in $S_C$.

**Generating captions and images.** For generating additional captions for the training images, we rely on GPT-3.5-turbo. For each input image, we provide the representation produced by our trained OpenCLIP model and ask GPT to generate five new captions. Generated sample captions are presented in Figure 18. To generate additional images for the COCO dataset, we use Stable Diffusion v1.5 to generate five new images, one corresponding to each of the five per-image captions in the COCO dataset. Sample generated images are presented in Figure 17.

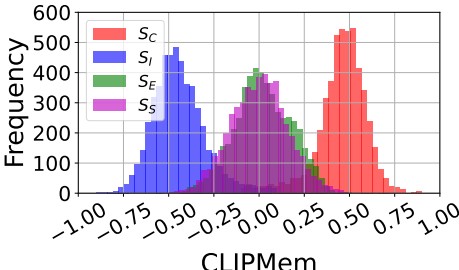

(a) Memorization scores across data subsets.

|  | Clean $S_C$ | Poisoned $S_C$ |
|---|---|---|
| Clean Model | 0.438 | N/A |
| Poisoned Model | 0.440 | 0.586 |

(b) Average CLIPMem scores.

Figure 2: **Memorization with CLIPMem**. We train a CLIP model on COCO using standard image cropping and no text augmentations. (a) We present memorization scores according to CLIPMem per data subset. The significantly higher scores for $S_C$ compared to $S_S$ indicate that $f$ memorizes $S_C$. (b) We also study how inserting training samples with imprecise or incorrect captions ("mis-captioned") affects memorization. We refer to the model trained with correct captions as **Clean Model**, and the model trained with $S_C$ containing 4500 standard canaries (**Clean**) and 500 mis-captioned (**Mis-captioned**) as **Poisoned Model**. We report CLIPMem over the different subsets of candidates. We observe that the mis-captioned samples experience a significantly higher memorization while the memorization of the clean data points is (almost) not affected.

## 4.2 STUDYING MEMORIZATION USING CLIPMEM

We first set out to analyze the general memorization in CLIP in order to identify which data points are memorized. To do this, we quantify CLIPMem over the different training subsets. Our results are presented in Figure 2a. In particular, we observe that CLIPMem for $S_C$, the data points only used to train model $f$, is significantly higher than for $S_S$, the data points shared between the two models. Memorization for $S_S$ is comparable to that for $S_E$, *i.e.,* the external data not seen during training, indicating that $f$ does not memorize these samples. The data in $S_I$ causes negative CLIPMem scores, indicating that this data is memorized by $g$, not by $f$. This is the expected behavior according to the definition of our metric. In Appendix A.4, we additionally highlight that memorization increases with model size, *i.e.,* CLIP based on ViT-Large has a higher overall memorization with an average of $0.457$ while CLIP based on ViT-Base only reaches $0.438$ on average.

Additionally, we analyze individual data points according to their reported CLIPMem. We give examples of highly memorized data points in CLIP in Figure 1 and more highly vs. little memorized samples in Figures 13,14,15, and 16 in Appendix A.9. Overall, the samples with high CLIPMem, *e.g.,* in Figure 1 seem to be difficult examples and examples with imprecise or incorrect captions whereas the samples with low CLIPMem are simpler and (potentially consequently) more precisely captioned. In Appendix A.5, we show that these findings also hold when we operate in the *infinite data regime*, *i.e.,* when we perform only a single training run where no data point is repeated.

Motivated by this insight and by observations from supervised learning where it was shown that models can memorize random labels (Zhang et al., 2016) and where mislabeled data experiences highest memorization (Feldman, 2020), we test if the same effect can also be observed in CLIP. Therefore, we "poison" our CLIP's training data by randomly shuffling the captions among 500 of the 5000 candidate data points in $S_C$. Thereby, these 500 data points are "mis-captioned". We train a model based on this data and see that the mis-captioned examples experience significantly higher memorization (CLIPMem of 0.586) compared to the "clean" data points (CLIPMem of 0.440). Even though CLIP trains using a contrastive training objective, the memorization of clean data points is not significantly affected by training the model with the mis-captioned examples, as we can see by their CLIPMem that is 0.438 on the clean model and 0.440 on the poisoned model.

## 4.3 MEASURING MEMORIZATION IN ONE MODALITY DOES NOT YIELD A STRONG SIGNAL

To understand how important it is to take both modalities into account in our definition of CLIPMem, we set out to evaluate whether existing practical methods to measure memorization over uni-modal encoders (Wang et al., 2024b) yield a sufficiently strong memorization signal in CLIP. Therefore, we

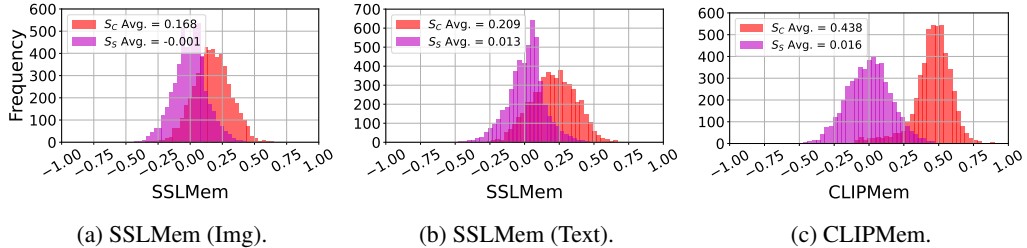

(a) SSLMem (Img).          (b) SSLMem (Text).          (c) CLIPMem.

Figure 3: **Measuring memorization on individual modalities is not able to extract a strong signal.** (a)–(b) We measure SSLMem (Wang et al., 2024b) on the individual encoders of our CLIP model trained on COCO. (c) Our CLIPMem extracts a stronger memorization signal by using both modalities in CLIP jointly.

apply their SSLMem to the individual encoder parts of CLIP. Since SSLMem relies on augmentations of the encoder input, we use image crops, like during CLIP training for the vision encoder, and the 5 COCO captions as augmentations for the text, like in (Fan et al., 2023). Our results in Figure 3 highlight that SSLMem and its naive adaptation to CLIP fail to yield a strong signal for memorization. In particular, there is a high overlap in scores between the non-memorized samples from $S_S$, and candidate examples for memorization $S_C$. Additionally, the highest reported memorization scores for $S_C$ go up to around 0.65 (for SSLMem on the vision encoder) and 0.73 (for SSLMem on the text encoder). In contrast, our new CLIPMem is able to get a distinct signal for the candidates $S_C$ with respect to $S_S$ and reports a much higher memorization of 0.91. Thereby, our CLIPMem prevents under-reporting the actual memorization in CLIP.

## 4.4 MEMORIZATION BETWEEN MODALITIES

Our results in Figure 3 indicate that memorization is higher in CLIP's text encoder than in the image encoder (the average SSLMem on $S_C$ in the text encoder is 0.209 vs. 0.168 in the image encoder). To provide further insights into how memorization behaves between the modalities in CLIP, we first analyze the use of augmentations. We compare five cases: (1) no additional augmentations beyond the baseline (image cropping), (2) generating one image using a diffusion model for a given original caption, (3) generating five variations of each image using a diffusion model and randomly selecting one for each training iteration while keeping the caption fixed, (4) using the original image but randomly selecting one of the five COCO captions for each training iteration, and (5) randomly pairing each of the five generated images with one of the five COCO captions.

As shown in Figure 14, there is quite a variability in the COCO captions for the same sample. Hence, some images might not fit well with the chosen training caption. This imprecise captioning can cause an increase in memorization. We observe that the effect is mitigated when using the 5 images with the 5 captions (5th case, see

Table 1: Impact of augmentations.

| Case | CLIPMem | Lin. Prob. Acc. (ImageNet) |
|---|---|---|
| 1 Image, 1 Caption | 0.438 | 63.11% ± 0.91% |
| 1 Image (generated), 1 Caption | 0.428 | 63.97% ± 0.79% |
| 5 Images (generated), 1 Caption | 0.424 | 64.60% ± 0.82% |
| 1 Image, 5 Captions | 0.423 | 64.88% ± 0.83% |
| 5 Images (generated), 5 Captions | 0.417 | 64.79% ± 0.99% |

Table 1). This phenomenon results most likely from the increased number of possible image-text pairs (25), such that individual incorrect or imprecise pairs are not seen so often during training. For the third case, *i.e.,* row three in Table 1, we generate five images with a diffusion model based on all five captions per image from the COCO dataset. However, as we only use the first caption during training, this would introduce many mis-captioned images which significantly lowers performance and increases memorization. To avoid this problem, we removed 6000 mis-captioned samples.

Our results in Table 1 highlight that augmenting text during training reduces memorization and increases performance more than augmenting images. However, applying augmentations of both text and images strikes the right balance between the reduction in memorization and the increase in performance. In fact, applying both augmentations reduces memorization most significantly. Overall, these results indicate that memorization in CLIP's is tightly coupled to the captions assigned to the training images with imprecise captions having a destructive effect on CLIP performance and memorization.

## 4.5 RELATION TO CLIP MEMORIZATION TO (SELF-)SUPERVISED MEMORIZATION

We further provide insights on whether CLIP's memorization behavior is more alike to the one of supervised learning or SSL. This question is highly interesting since the captions in CLIP can be considered as a form of labels, like in supervised learning, whereas the contrastive training objective on the dataset resembles more SSL. We perform two experiments to gain a better understanding of the memorization behavior of CLIP with respect to supervised learning and SSL.

First, we compare an SSL vision encoder pair $f$ and $g$ with the same architecture as CLIP's vision encoder but trained from scratch on COCO using DINO, *i.e.,* standard SSL training. We train $f$ and $g$ using the same candidates as the pair of CLIP models in our previous experiments. Then, we use the SSLMem metric from Wang et al. (2024b) to quantify memorization in the CLIP vision encoder and the SSL encoder, respectively. The CLIP vision encoder has a significantly lower SSLMem than the SSL encoder (0.209 vs. 0.279). Hence, CLIP vision encoders experience lower SSL memorization than SSL trained encoders. To further investigate the difference, we also report the overlap between the top 10% memorized samples between the two models, measured according to SSLMem. With an overlap of only 47 out of 500 (9.4%) samples, we find that CLIP memorizes significantly different samples than SSL encoders. Wang et al. (2024b) had performed a similar experiment on SSL vs. supervised learning and found that the two paradigms also lead to different samples being memorized. While this is, on the one hand, an effect of the different objective function, the difference between the memorized samples in CLIP and SSL is likely also closely connected to the additional captions that CLIP takes into account. While SSL-trained encoders can memorize atypical images, CLIP encoders can memorize typical images when they have an atypical, imprecise, or incorrect caption.

Additionally, we compare the memorization behavior of CLIP against supervised and SSL-trained models on the neuron-level. Therefore, we train two additional ViT-Base models on COCO using supervised training and SSL training with DINO. Then, we apply the UnitMem metric (Wang et al., 2024a) to measures how much individual neurons memorize individual samples from the training data. A high UnitMem suggests that neurons highly memorize individual data points instead of groups/classes of points. It had been shown that supervised learning causes neurons in lower layers to experience low UnitMem, *i.e.,* being responsible for learning joint groups of data points,

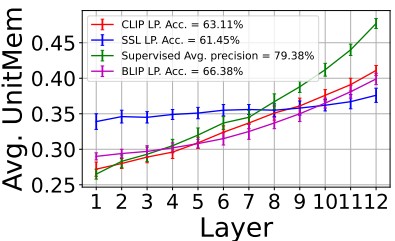

Figure 4: **UnitMem metric: CLIP is between supervised and SSL models.**

while neurons in later layers highly memorize individual data points. In contrast, for SSL, UnitMem was shown to remain relatively constant over layers with neurons in lower layers also being able to memorize individual data points. This difference was attributed to the different objective functions where supervised learning's cross entropy loss pulls together data points from the same class, whereas SSL's contrastive loss leads to individual data points being pushed away from each other (Wang et al., 2024a). Our results in Figure 4 highlight that CLIP, in terms of its memorization behavior, is between supervised learning and SSL. At the lower layers, it is much less selective than models trained with SSL, *i.e.,* it focuses on groups of data points rather than memorizing individual data points, similar to supervised learning. Yet, in later layers, CLIP becomes more selective than SSL, *i.e.,* it memorizes individual data points more in individual neurons, but still less than supervised learning which there has a very high average per-layer UnitMem.

## 4.6 MITIGATING MEMORIZATION WHILE MAINTAINING GENERALIZATION

The experiments from Table 1 suggest that using augmentations during training can improve generalization while also reducing memorization. This is an unexpected synergy since for both supervised learning (Feldman, 2020) and SSL (Wang et al., 2024b), generalization was shown to decline when memorization decreases. To further study the impact of mitigating memorization in CLIP on downstream generalization, we explore two orthogonal strategies for "augmenting" the text modality during CLIP training, first in the input space and second directly in the embedding space. Additionally, we analyze the effect of removing memorized samples from training.

**Multiple captions.** We vary the number of captions used during training and report fine-grained insights into the resulting memorization and downstream performance in Figure 5a. Our results

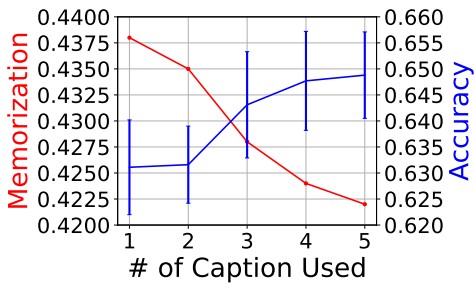

(a) Different numbers of captions.

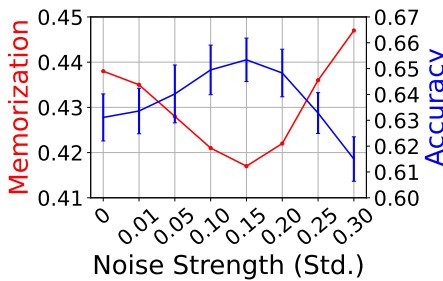

(b) Noising text embedding during training.

Figure 5: **Mitigating memorization in CLIP improves downstream generalization.** We train CLIP models with different "augmentations" in the textual domain. (a) We use multiple captions for the same image during training. (b) We directly noise the text embeddings during the training using Gaussian noise with a mean of 0 and different standard deviations (adding the Gaussian noise $\mathcal{N}(0, 0.15)$ gives us the sweet spot with the smallest memorization and highest performance). Both strategies successfully reduce memorization while improving performance.

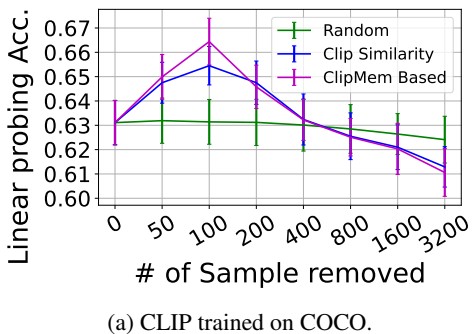

(a) CLIP trained on COCO.

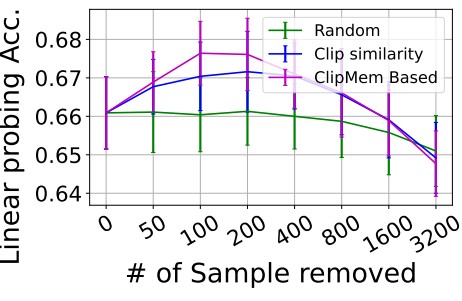

(b) CLIP trained on CC3M.

Figure 6: **Removing memorized samples according to CLIPMem has a stronger influence on the linear probing accuracy than removing random data points.** Removing the mislabeled samples based on CLIPMem improves the performance significantly, followed by a sharper drop when removing atypical samples.

highlight the trend that the more captions are used during training, the lower memorization and the higher the linear probing accuracy. Our additional results in Table 3 highlight also that choosing all captions equally often is beneficial for utility while keeping memorization roughly the same. Since not in every dataset, multiple captions are available, we experiment with generating these captions with a language model. Our results in Table 7 where we train CLIP with captions generated by GPT3.5 show that the results both in terms of utility and memorization are extremely similar to the original captions, making this improved training strategy widely applicable. Our findings that modifying the text during training can reduce memorization align with the insights presented by Jayaraman et al. (2024). For datasets where only single captions are available, they proposed *text randomization*, *i.e.,* masking out a fraction of tokens during training as a mitigation for their Déjà Vu memorization. In contrast to our GPT3.5-generated captions, this masking, however, causes a drop in performance when mitigating memorization. We hypothesize that this is due to the higher distribution shift introduced by the masked tokens.

**Noising the text embedding during training.** To overcome such shortcomings altogether and avoid any inherent distribution shifts, we propose to perform the "augmentations" directly in the embedding space. More precisely, we experiment with an approach where, during training, before calculating the cosine similarity between text and image embeddings for the contrastive loss, we add small amounts of Gaussian noise to the text embeddings. Our results in Figure 5b and Table 8 highlight that this strategy is highly effective in reducing memorization while improving downstream generalization.

**Removing memorized samples.** Finally, we investigate the effect of removing memorized samples to understand how it impacts downstream performance. We perform an additional experiment where

we first train a CLIP model, then identify the highest memorized training data points, remove them, and retrain on the remaining data points only. We compare this method to two baselines where we either randomly remove samples or filter out the samples with the lowest CLIP similarity between the training data points' two modalities. We showcase the effect on the downstream linear probing accuracy on ImageNet in Figure 6 with CLIP models trained on COCO and on the CCM3 dataset. For the COCO dataset, when removing up to 100 most memorized data points, we first observe a sharp increase in downstream performance in comparison to removing random samples. Then, the downstream performance starts dropping significantly more when removing memorized instead of random samples, until between 400 and 800 removed samples, the cutoff point is reached where model performance is worse when removing according to highest memorization instead of randomly. For the CC3M dataset, this cutoff occurs later, between 1600 and 3200 removed samples. While the CLIP similarity also manages to increase performance through removal, it is not as effective as CLIPMem, highlighting the value of considering memorization as a lens to identify noisy samples. This finding is significantly different than for supervised learning and SSL, where the removal of highly memorized samples *constantly* harms performance more than the removal of random samples (Feldman, 2020; Wang et al., 2024b). We hypothesize that the effect observed in CLIP might result from the distinction between "mis-captioned" and atypical samples, where the former harm generalization while the latter help the model learn from smaller sub-populations (Feldman, 2020). We empirically support this hypothesis in Appendix A.3.1. The finding that CLIP generalization can be improved by identifying inaccurately captioned data points using our CLIPMem and removing them from training is of high practical impact, given that state-of-the-art CLIP models are usually trained on large, uncurated datasets sourced from the internet with no guarantees regarding the correctness of the text-image pairs. Overall, our results suggest that CLIPMem can help reduce memorization in CLIP while improving downstream generalization.

## 5 CONCLUSION

We presented CLIPMem, a formal measure to capture memorization in multi-modal models, such as CLIP. By not only quantifying memorization but also identifying *which* data points are memorized and *why*, we provide deeper insights into the underlying mechanisms of CLIP. Our findings highlight that memorization behavior of CLIP models falls between that of supervised and self-supervised models. In particular, CLIP highly memorizes data points with incorrect and imprecise captions, much like supervised models memorize mislabeled samples, but it also memorizes atypical examples. Furthermore, we find that memorization in CLIP happens mainly within the text encoder, which motivates instantiating mitigation strategies there. By doing so, we can not only *reduce memorization* in CLIP but also *improve* downstream generalization, a result that challenges the typical trade-offs seen in both supervised and self-supervised learning.

## ACKNOWLEDGEMENTS

This research was funded by the Deutsche Forschungsgemeinschaft (DFG, German Research Foundation), Project number 550224287. We would like to also acknowledge our sponsors, who support our research with financial and in-kind contributions, especially the OpenAI Cybersecurity Grant.

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

# A APPENDIX

## A.1 EXTENDED BACKGROUND

**Déjà Vu Memorization in CLIP.** The Déjà Vu memorization framework (Jayaraman et al., 2024) is the only existing other work that attempts to quantify memorization in vision-language models. It uses the text embedding of a training image caption to retrieve relevant images from a public dataset of images. It then measures the fraction of ground-truth objects from the original image that are present in the retrieved images. If the training pair is memorized, retrieved images have a higher overlap in ground truth objects, beyond the simple correlation. While valuable, several aspects warrant further consideration for broader applicability of the framework. First, its focus on object-level memorization ignores non-object information like spatial relationships or visual patterns that can also influence memorization (Feldman, 2020; Wang et al., 2024b). To perform object retrieval, the framework also relies on object detection and annotation tools, which may introduce variability based on the accuracy and robustness of these tools. Additionally, the assumption that public datasets with similar distributions to the training data are readily available may not always hold, necessitating alternative approaches. Moreover, the framework does not analyze why certain images are memorized limiting detailed analysis. Finally, while Déjà Vu must address the challenge of distinguishing between memorization and spurious correlations, CLIPMem avoids this by directly assessing memorization on the output representations of the model. One notable difference between the results of our approach and Déjà Vu's is that their findings show that their mitigation strategies can reduce memorization, but at the cost of decreased model utility. CLIPMem, in contrast, does not observe trade-offs between memorization and performance.

## A.2 EXTENDED EXPERIMENTAL SETUP

**General Setup.** All the experiments in the paper are done on a server with 4 A100 (80 GB) GPUs and a work station with one RTX 4090 GPU(24 GB). We detail the setup for our model training, both CLIP and SSL (relying on DINO) in Table 2.

Table 2: **Experimental Setup.** We provide details on our setup for encoder training and evaluation.

| | Model Training | | | Linear Probing | | |
|---|---|---|---|---|---|---|
| | CLIP | DINO | Supervised ViT | CLIP | DINO | Supervised ViT |
| Training Epoch | 100 | 300 | 100 | 45 | 45 | 45 |
| Warm-up Epoch | 5 | 30 | 5 | 5 | 5 | 5 |
| Batch Size | 128 | 1024 | 128 | 4096 | 4096 | 4096 |
| Optimizer | Adam | AdamW | Adam | LARS | LARS | LARS |
| Learning rate | 1.2e-3 | 2e-3 | 1e-3 | 1.6 | 1.6 | 1.6 |
| Learning rate Schedule | Cos. Decay | Cos. Decay | Cos. Decay | Cos. Decay | Cos. Decay | Cos. Decay |

**Experimental Setup for SSLMem.** To experimentally evaluate memorization using the SSLMem framework (Wang et al., 2024b), the training dataset $S$ is split into four sets: *shared set* ($S_S$) used for training both encoders $f$ and $g$; *candidate set* ($S_C$) used only for training encoder $f$; *independent set* ($S_I$) data used only for training encoder $g$; and an additional *extra set* ($S_I$) from the test set not used for training either $f$ or $g$. For training encoders, encoder $f$ is trained on $S_S \cup S_C$, while encoder $g$ is trained on $S_S \cup S_I$. The alignment losses $\mathcal{L}_{\text{align}}(f, x)$ and $\mathcal{L}_{\text{align}}(g, x)$ are computed for both encoders, and the memorization score $m(x)$ for each data point is derived as the difference between these alignment losses, normalized to a range between $-1$ and $1$. A score of $0$ indicates no memorization, $+1$ indicates the strongest memorization by $f$, and $-1$ indicates the strongest memorization by $g$.

**Normalization on CLIPMem.** For improved interpretability, we normalize our CLIPMem scores to a range of $[-1, 1]$. A memorization score of $0$ indicates no memorization, $+1$ indicates the strongest memorization on CLIP model f, and $-1$ indicates the strongest memorization on CLIP model g. We find the normalized CLIPMem score for a dataset using the following process: For each image-text pair $(I, T)$, we first calculate the CLIPMem score as the difference in alignment scores between two CLIP models $f$ and $g$. Once CLIPMem scores are computed for all data points, we normalize them by dividing each score by the range, which is the difference between the maximum and minimum scores in the dataset. Finally, we report the normalized CLIPMem score for a dataset as the average of these normalized values.

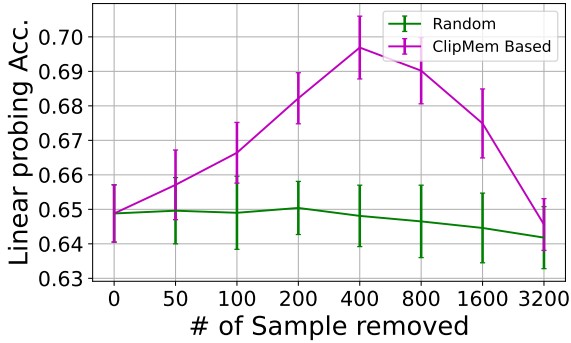

Figure 7: **Removing memorized samples.** We show the effect on downstream performance in terms of ImageNet linear probing accuracy and CLIPMem for a CLIP model trained on COCO using 5 text captions instead of 1, like done in Figure 6. We observe the same trend, with the difference that the peak is at roughly 500 removed samples rather than 100. This is likely due to the increase in captions (by factor 5) that causes increase in mis-captioned samples.

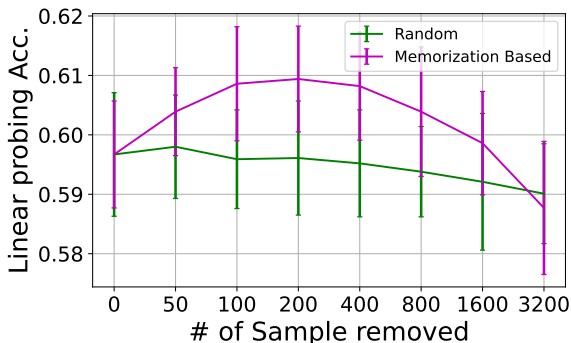

Figure 8: **Removing memorized samples in supervised learning.** We train a ViT-tiny on CI-FAR10 (Krizhevsky et al., 2009) using supervised learning. We use our evaluation setup with $S_C$, $S_S$, $S_I$, and $S_E$ to approximate the memorization metric from Feldman (2020). We use 5000 samples in $S_C$, but before training, we flip the labels of 200 samples. We calculate memorization over all samples in $S_C$ and test the linear probing accuracy with ImageNet resized to 32*32 on the representations output before the original classification layer.

## A.3 ADDITIONAL EXPERIMENTS

### A.3.1 MEMORIZATION VS. GENERALIZATION IN CLIP

**Extending evaluation.** In Figure 7, we perform the same experiment as in Figure 6, but on a CLIP model trained with 5 captions instead of 1. We observe the same trend, with the difference that the peak is at roughly 500 removed samples rather than 100. This is likely due to the increase in captions (by factor 5) that causes increase in mis-captioned samples.

**Verifying the hypothesis on memorizing mis-captioned samples through supervised learning.** We repeat the same experiment in the supervised learning setup to understand where the increase and then decrease in linear probing accuracy stems from. To test our hypothesis that it stems from "mis-captioned" samples, we "poison" our supervised model by flipping the labels of 200 data points before training. Then, we approximate the memorization metric from Feldman (2020) in our setup and remove highly memorized vs. random data points. In the same vein as in Appendix A.3.1, we first observe an increase in linear probing accuracy when removing memorized samples (instead of random samples). The peak is at roughly 200 data points, *i.e.*, the number of deliberately mislabeled samples. Until the cutoff point at roughly 3200 examples, linear probing accuracy is still higher when removing most memorized rather than random samples, which might suggest that there are other

Table 3: **Using different/multiple captions during training.** We evaluate CLIPMem how memorization on different data subsets and linear probing accuracy on ImageNet differ when using 1 caption (baseline), 5 COCO captions, one chosen at random at every round (random), and 5 COCO captions, but all chosen equally often, *i.e.,* 20 out of 100 training epochs (balanced). We observe that increasing the number of captions reduces highest memorization. Yet, only when we balance the usage of caption, also model performance increases.

|  | baseline | random | balanced |
|---|---|---|---|
| Avg. CLIPMem (Top 10 samples) | 0.792 | 0.788 | 0.790 |
| Avg. CLIPMem (Top 20%) | 0.552 | 0.531 | 0.540 |
| Linear Probing Acc. | 63.11% $\pm$ 0.91% | 62.44% $\pm$ 1.18% | 64.88% $\pm$ 0.83% |

Table 4: **The CLIPMem and linear probing accuracy of model trained with original coco captions and captions generated by GPT3.5.** For 'Single Caption', only one caption is used during training. For 'Five Caption', all five caption are used equally during training (every caption trained for 20 epoch out of 100). The linear probing accuracy is tested on ImageNet

|  | COCO | | GPT3.5 | |
|---|---|---|---|---|
|  | Single Caption | Five Caption | Single Caption | Five Caption |
| CLIPMem | 0.438 | 0.423 | 0.430 | 0.411 |
| LP. Acc. | 63.11% $\pm$ 0.91% | 64.88% $\pm$ 0.83% | 63.09% $\pm$ 1.12% | 64.47 $\pm$ 0.72% |

outliers or inherently mislabeled samples whose removal improves model performance. After the cutoff, we observe the behavior as observed in prior work (Wang et al., 2024b; Feldman, 2020) that reducing memorization harms generalization more than reducing random data points from training.

### A.3.2 THE EFFECT OF CAPTIONS

In Table 3, we show that using more captions during training reduces memorization and that by using each caption at the same frequency over the training epochs, we can additionally improve model performance. Additionally, we show that captions generated by GPT3.5 have the same effect as the original COCO captions on memorization and linear probing accuracy in Table 4.

### A.4 THE EFFECT OF MODEL SIZE

In Table 5, we present how the model size affects the memorization level of CLIP models. Both models are trained using the same dataset and settings. We observe that with more parameters (larger model size), encoders have higher memorization capacity. This aligns with findings from previous research (Wang et al., 2024b; Feldman, 2020; Meehan et al., 2023).

### A.5 VERIFICATION OF INFINITE DATA REGIMES

To evaluate CLIPMem over infinite data regimes (*i.e.,* using a single training run where no data point is repeated), we use a subset $D$ (containing 7050000 samples) of YFCC100M dataset (Thomee et al., 2016b) to train another pair of ViT-Base models for only 1 epoch. Following our definition of CLIPMem, we further divide $D$ into $S_S$ with 6950000 samples, $S_C$ with 50000 samples, and $S_I$ with 50000 samples. The reason we use 7M (6950000+50000) samples to train either model $f$ or model $g$ is to make sure the newly trained model has the same number of training samples as the model trained with K-epoch runs (70000 samples/epoch * 100 epoch). The results in Table 6 show that the model trained with infinite data regimes has higher linear probing accuracy on ImageNet as a downstream task and lower memorization scores, as measured by CLIPMem. This aligns with the fact that duplicated data points increase the memorization level and make the model over-fit, hence reducing the generalization (Wang et al., 2024b; Feldman, 2020). The results in Figure 9 show that the most memorized samples according to CLIPMem in the model trained with infinite data regimes are also samples with imprecise or incorrect captions. This aligns with our statements in Section 4.5.

Table 5: **CLIPMem and linear probing accuracy of models with different sizes.** The models are trained using identical settings and the same subset of the COCO dataset. Linear probing accuracy is tested on the ImageNet dataset as the downstream task.

| Model | CLIPMem | Lin. Prob. Acc. (ImageNet) |
|---|---|---|
| ViT-base (Baseline in main paper) | 0.438 | $63.11\% \pm 0.91\%$ |
| ViT-large | 0.457 | $67.04\% \pm 1.05\%$ |

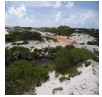 Series taken during heritage surveys in 2007. I have not had time to develop individual captions for these, however happy to provide some comment if required.

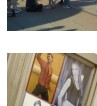 Photo By Terri Hodges www.craftingthegalaxy.com Useage is Creative Commons but credit is required

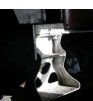 Cumbres and Toltec Scenic Railroad, historical narrow gauge railroad between Chama, New Mexico, and Antonito, Colorado. These photos are from the station and railyard in Chama.

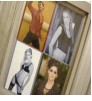 Mellwood Art Center in Louisville, KY (1-27-06)

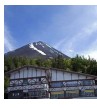 Day Four set here. Entire Japan collection here.

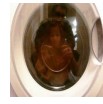 -- Lisa Brewster Sent from my Palm Pre

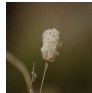 Natural Bridge & Rockshelter Wisconsin State Natural Area #105 Natural Bridge State Park Sauk County

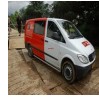 At Hellfest with After Forever

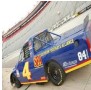 Mountain Empire Filed Trip - Bristol, TN, and environs Photo by Amy C. Evans, SFA oral historian June 2009 www.southernfoodways.org

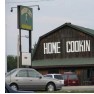 On the way to visit Western Kentucky University (9/20/08)

Figure 9: **Top 10 memorized samples according to CLIPMem in the model trained under infinite data regimes on YFCC100M.** The model is trained for one epoch, *i.e.,* seeing each training data point exactly once. Even in this setup, the most memorized samples are still the ones with imprecise or incorrect captions.

## A.6 EVALUATION ON BLIP

To verify the effectiveness of CLIPMem over other similar multi-modal models, we train a BLIP model on COCO dataset following the same settings as the baseline model in the main paper. We present the results for CLIPMem on BLIP over all four data subsets in Figure 10, which is in agreement with the results of the BLIP model in Figure 2a. We also present the results for UnitMem in Figure 4, which is also very similar to the results of CLIP models

## A.7 MEMORIZATION DISTRIBUTION DURING TRAINING

We present the distributions of neurons with highest UnitMem during training in Figure 11. These results highly consistently indicate that in the early stages of training, neuronal memory occurs mainly in the lower layer of the clip model, while in the middle and later stages of training, neuronal memory is more concentrated in the later layer of the model.

## A.8 HUMAN VS MACHINE GENERATED CAPTIONS

For each image in the COCO dataset, we use GPT 3.5 (specifically, gpt-3.5-turbo) to generate 5 captions (from scratch). We use the following instruction in the OpenAI API:

```
def generate_description_for_image(image_caption, clip_features):
    prompt = f"Here is an image with the caption: '{image_caption}'. "
    prompt += f"Based on this caption and the visual features
    represented by this embedding '{clip_features}',
    please generate a new detailed description."
```

Table 6: **Evaluation of CLIPMem under infinite data regimes, *i.e.,* seeing every data point only once during training vs training with 100 epochs.** We observe that both setups reach comparable downstream accuracy and memorization.

| Model | CLIPMem | Lin. Prob. Acc. (ImageNet) |
|---|---|---|
| ViT-Base (YFCC 7M, 1 epoch) | 0.425 | 64.83% ± 1.04% |
| ViT-Base (COCO 70K, 100 epochs) | 0.438 | 63.11% ± 0.91% |

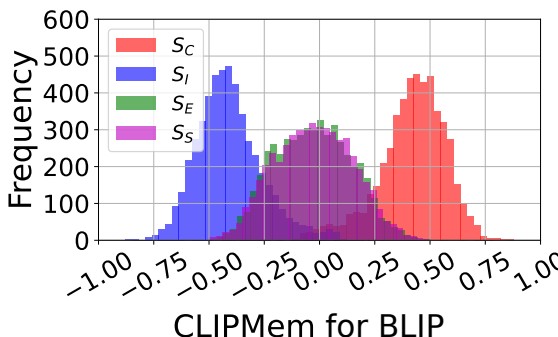

Figure 10: **Memorization scores across data subsets on BLIP models** We train a BLIP model on COCO standard image cropping and no text augmentation. We present the results for CLIPMem over all 4 data subsets, which is in agreement with the results of the CLIP model in Figure 2a

```
response = openai.ChatCompletion.create(
    model="gpt-3.5-turbo",
    messages=[
        {"role": "system", "content": "You are a helpful
        assistant that generates captions for images."},
        {"role": "user", "content": prompt}
    ]
)
return response['choices'][0]['message']['content']
```

We present the obtained captions in Figure 18. In Figure 12b, we analyze the pairwise cosine similarity in the original COCO and the GPT3.5 generated captions. We find that the GPT3.5 generated captions are slightly more uniform than the original COCO captions, reflecting in a higher pairwise cosine similarity.

## A.9 EXAMPLES FOR MEMORIZED SAMPLES

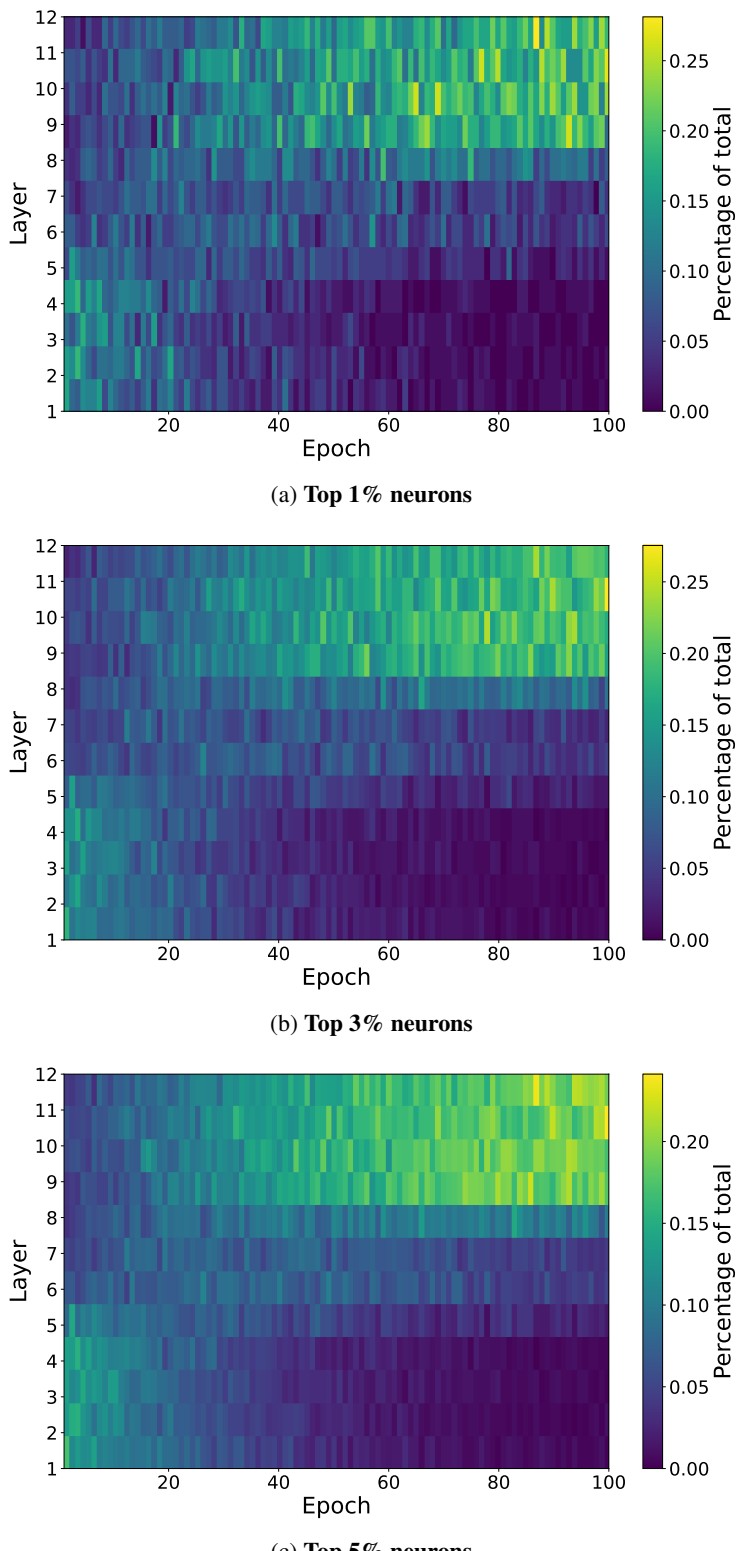

(a) **Top 1% neurons**

(b) **Top 3% neurons**

(c) **Top 5% neurons**

Figure 11: **Distribution of top 1%, 3%, and 5% neurons with highest UnitMem during training.**
We train a CLIP model on COCO standard image cropping and no text augmentation following the
settings of baseline model in main paper. We record the neurons with top 1%, 3%, and 5% of highest
UnitMem during training (every epoch).

Table 7: **The machine generated captions provide similar performance to the original human-generated captions.** We report the CLIPMem and linear probing accuracy of model trained with original COCO captions and captions generated by GPT 3.5. For the 'Single Caption', only a single caption is used during training. For 'Five Captions', all five captions are used equally during training (every caption trained for 20 epochs out of 100). The linear probing accuracy is tested on the ImageNet dataset as the downstream task.

| | COCO | | GPT 3.5 | |
| | Single Caption | Five Captions | Single Caption | Five Captions |
| --- | --- | --- | --- | --- |
| CLIPMem | 0.438 | 0.423 | 0.430 | 0.411 |
| Linear Probing Accuracy (ImageNet) | 63.11% ± 0.91% | 64.88% ± 0.83% | 63.09% ± 1.12% | 64.47% ± 0.72% |

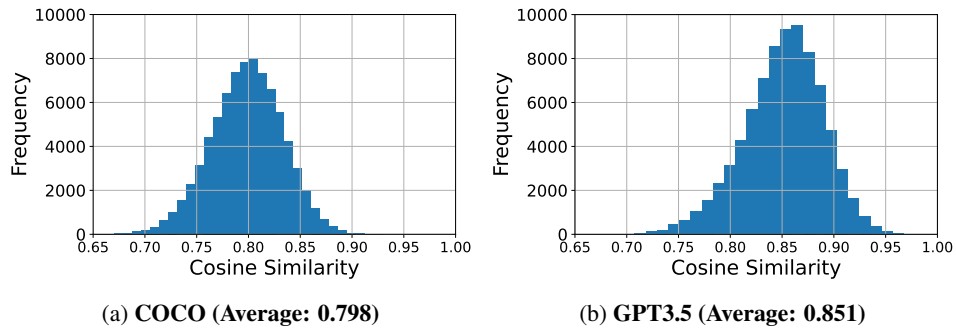

(a) **COCO (Average: 0.798)**    (b) **GPT3.5 (Average: 0.851)**

Figure 12: **Pairwise cosine similarity of 5 captions from COCO and generated by GPT3.5.**

| Noise | CLIPMem | Lin. Prob. Acc. (ImageNet) |
| --- | --- | --- |
| None | 0.438 | 63.11% ± 0.91% |
| $\mathcal{N}(0.01)$ | 0.435 | 63.36% ± 0.88% |
| $\mathcal{N}(0.05)$ | 0.428 | 64.02% ± 1.12% |
| $\mathcal{N}(0.10)$ | 0.421 | 64.95% ± 0.96% |
| $\mathbf{\mathcal{N}(0.15)}$ | **0.417** | **65.34% ± 0.84%** |
| $\mathcal{N}(0.20)$ | 0.422 | 64.83% ± 0.92% |
| $\mathcal{N}(0.25)$ | 0.436 | 63.28% ± 0.79% |
| $\mathcal{N}(0.30)$ | 0.447 | 61.50% ± 0.86% |
| $\mathcal{N}(0.50)$ | 0.491 | 57.04% ± 1.11% |
| $\mathcal{N}(0.75)$ | 0.501 | 52.28% ± 0.98% |
| $\mathcal{N}(1.00)$ | 0.504 | 51.92% ± 1.03% |

Table 8: **Noising text embedding during training.** We present the impact of adding noise to the text embedding during training for the ViT-base trained on COCO.

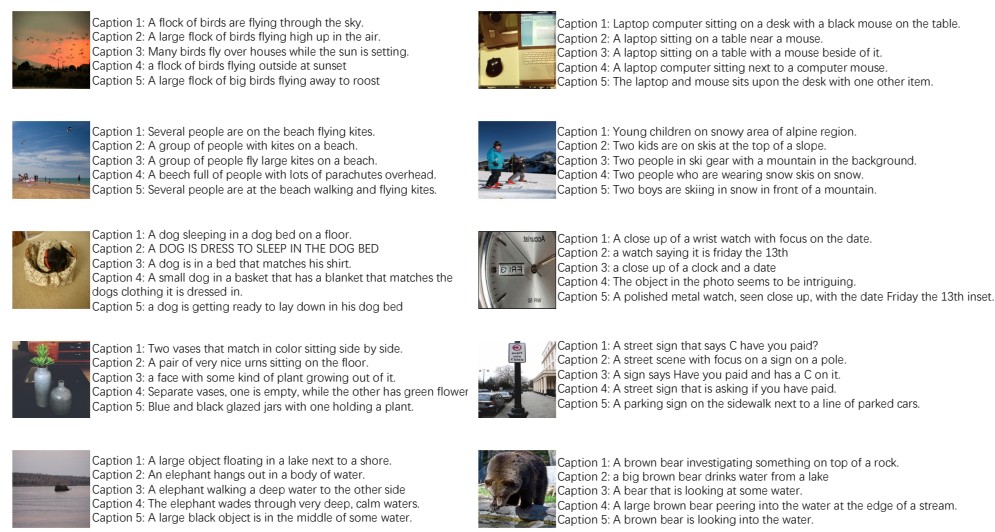

**Figure 13: The 10 samples with lowest CLIPMem in the CLIP model trained with all 5 captions.** We can see that these samples contain clear concepts and precise captions.

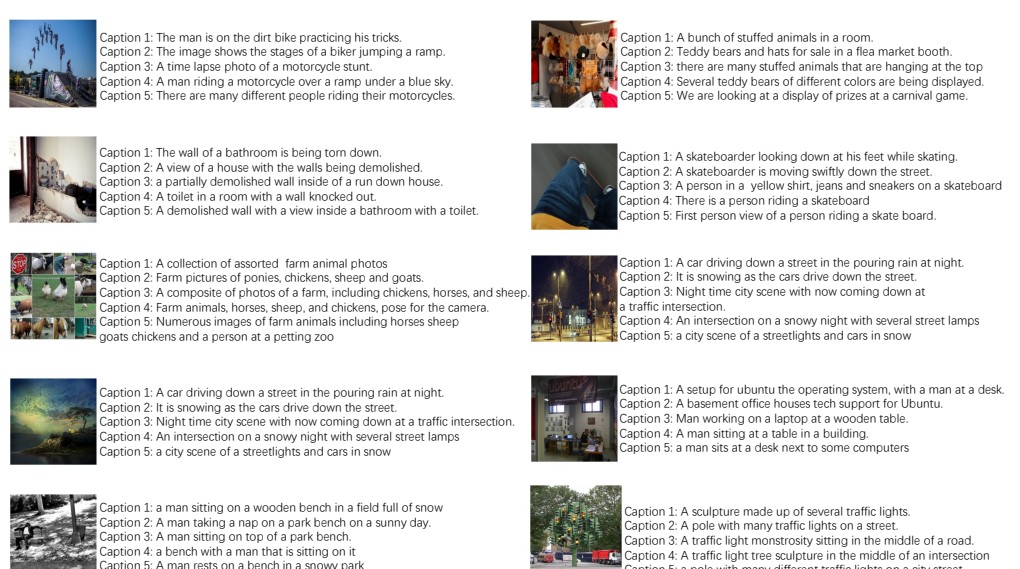

**Figure 14: The 10 samples with highest CLIPMem in the CLIP model trained with all 5 captions.** We can see that these samples contain atypical, difficult samples with imprecise or incorrect captions.

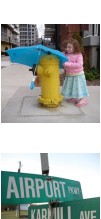 A girl in pink sweater putting a blue umbrella over a yellow fire hydrant.

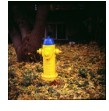 A yellow and blue fire hydrant surrounded by leaves

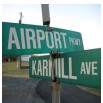 Closeup of two street signs that read "Airport Pkwy" and "Karmill Ave."

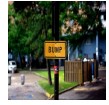 A sign is displayed on a pole that says bump.

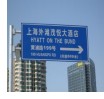 A sign with Oriental writing and the words saying Hyatt on the Bund.

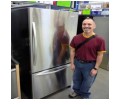 A man in maroon shirt standing next to a stainless steel refrigerator.

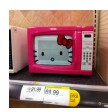 A pink Hello Kitty microwave on a store shelf.

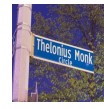 A blue street sign that reads "Thelonius Monk Circle."

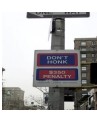 A sign saying "Don't Honk, $350 Penalty" on a pole.

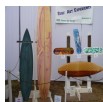 A group of artistic surfboards are displayed in a tent.

Figure 15: **The 10 samples with lowest CLIPMem in the CLIP model trained with 1 caption.** We can see that these samples contain clear concepts and precise captions.

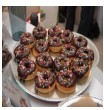 An item of a vase with something inside of it.

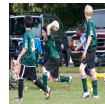 A group of people standing on top of a field together.

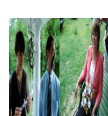 two males and a female in a red top holding some flowers

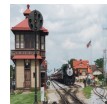 a pole with some yellow lights in front of a narrow building

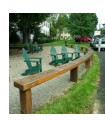 A woman opening the trunk of her car.

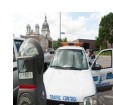 a parking lot with a bunch of cars in it

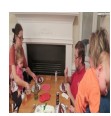 a sleeping toddler laying on a womans shoulder

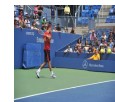 A person is taken in this very picture.

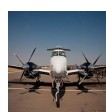 An open point of view of a room with various things all around

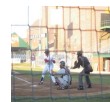 I am unable to see an image above

Figure 16: **The 10 samples with highest CLIPMem in the CLIP model trained with 1 caption.** We can see that these samples contain atypical, difficult samples with imprecise or incorrect captions.

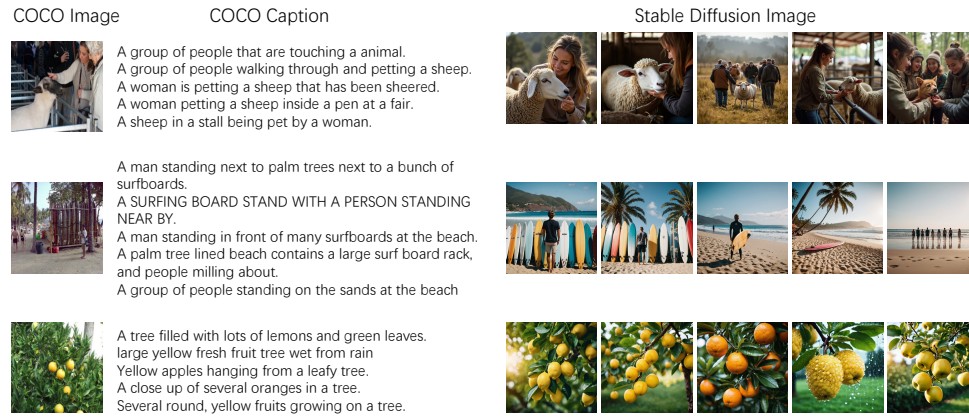

Figure 17: **Samples of images generated by Stable Diffusion.** We present the generated images based on the COCO captions.

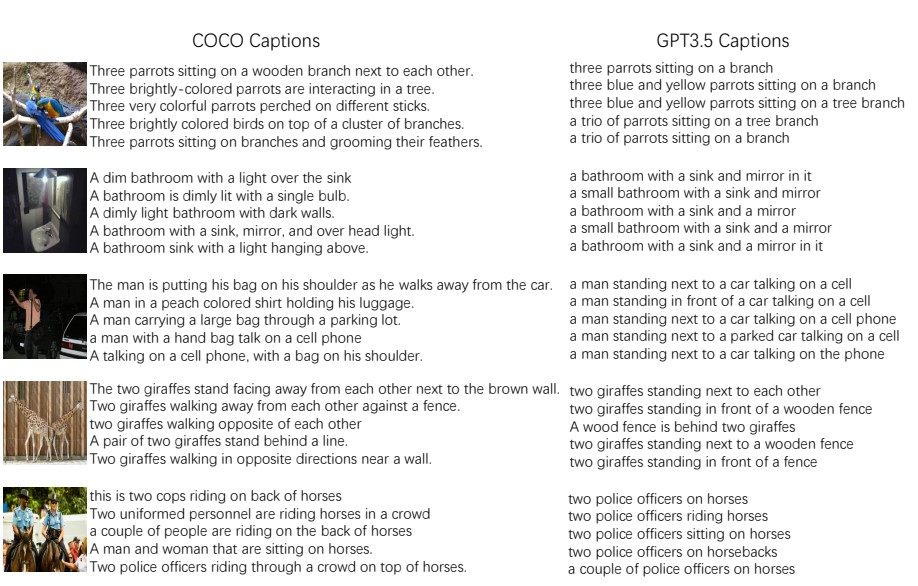

Figure 18: **Sample captions generated by GPT3.5.** We present the generated captions and the original image and captions from COCO.

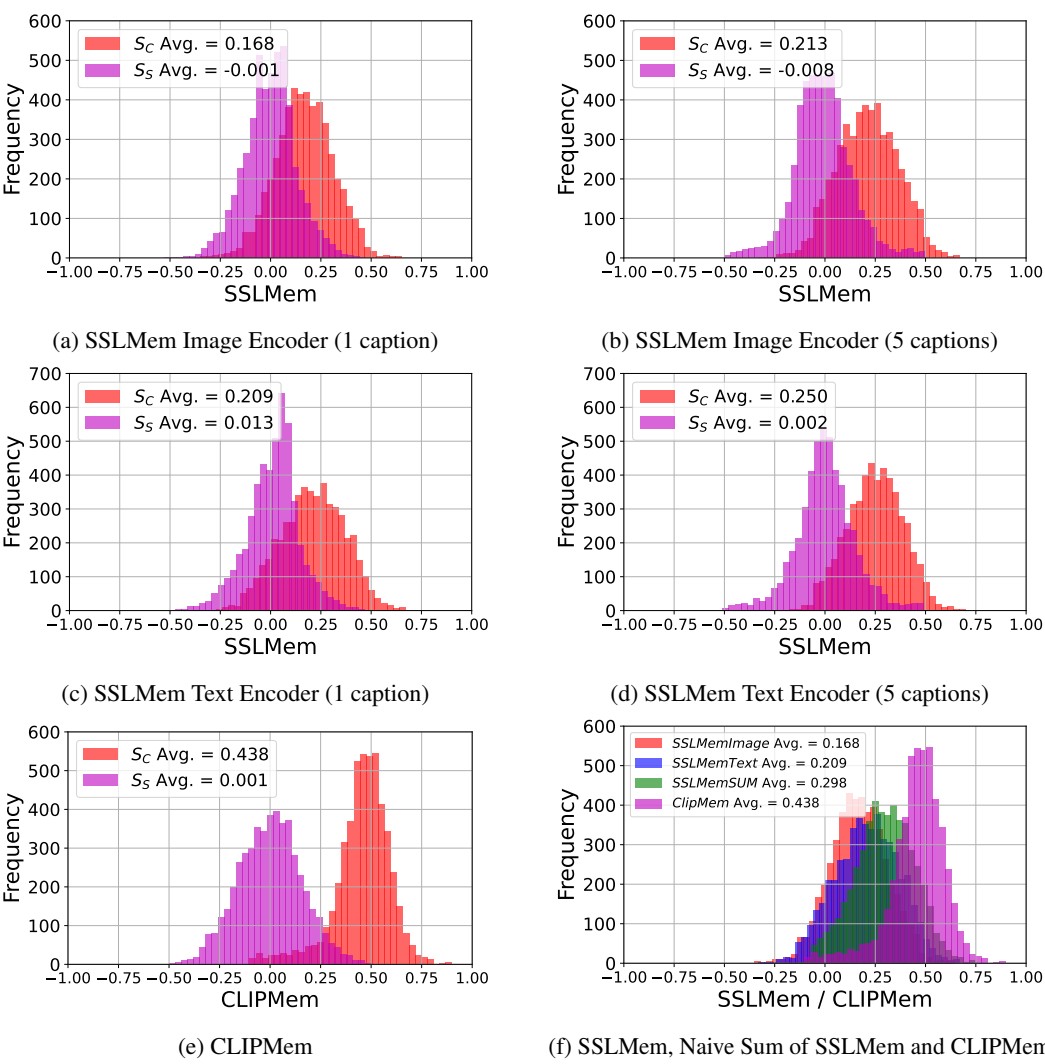

Figure 19: **Evaluation of SSLMem and CLIPMem on a CLIP model trained on COCO.** Extended version of Figure 3 where we also include SSLMem calculated on encoders trained with 5 captions instead of 1. The trends in both cases are the same. SSLMem for the CLIP Models trained with the 5 captions is slightly higher since SSLMem uses the captions as augmentations for the calculation of the memorization. Overall, our CLIPMem reports the strongest memorization signal for CLIP.

