# OpenReview forum: "Captured by Captions: On Memorization and its Mitigation in CLIP Models"
_ICLR.cc/2025/Conference — ICLR 2025 Poster_

### Official Review · Reviewer_F6JJ · 2024-10-25

**Soundness:** 3
**Presentation:** 3
**Contribution:** 2
**Rating:** 5
**Confidence:** 4

**Summary:**

They study the memorization problem in the CLIP model unlike existing studies focusing on the unimodal memorization problem.
They propose a new metric to analyze it and find that memorization seems to be more significant in text encoder than in image encoder. Their analysis indicates that augmenting captions can be a key to mitigating memorization in the CLIP model. Their experiments confirm that augmenting captions can improve the quality of the image encoder's representations while reducing memorization.

**Strengths:**

1. They propose a new metric to measure memorization in the CLIP model. The design of the metric is reasonable.
2. Their insight that memorization is more significant in the text encoder is new and might interest readers.
3. They conduct analysis on the augmentation of the text and images, which might be also interesting to some readers.
4. This paper is well-organized and easy to follow.

**Weaknesses:**

1. They do not discuss how model size can affect memorization. Although I am not very familiar with this topic, I guess the model size can affect their arguments. For example, if they utilize a larger image encoder, the memorization might be more significant on the image side. Therefore, I think their conclusion about which encoders suffer more from memorization can change by the size of the encoders, but they do not discuss much.

2. Most of their findings sound a bit too reasonable and are not surprising. Their finding that augmenting text improves the CLIP model has already been observed in many previous papers though they probably did not discuss memorization. Also, it is not hard to imagine that augmenting datasets mitigates memorization. In these points, I think their findings are not impressive.

3. They mention that their metric is effective in removing noisy samples. But, they compare their approach only with random replacement. I think they need to add more baseline, such as naive CLIP's similarity as done in many works.

4. In Table 1, the authors augment images by using a diffusion model. But, as they imply, such augmentation can cause a distribution shift in the image side and does not give much intuition about image augmentation.

Overall, I think this paper is well-organized and delivers a clear statement to readers, which I like. However, I think their findings are not very surprising and lack impact due to the reasons described in 1, 2. My rating is based on it.

**Questions:**

My rating is mainly based on 1 and 2. Please respond to those points.

---

> ### Author Response · Authors · 2024-11-24
> **Model size vs Memorization, Surprising findings, CLIPMem vs CLIP Similarity, and Augmentations**
>
> We thank the reviewer for recognizing that our metric is reasonable and that the insights on memorization provided are of interest to the readers.
>
> >**They do not discuss how model size can affect memorization.**
>
> To address the reviewer’s comments, we conducted additional experiments where we varied the size of the encoders. Note that since CLIP embeds both text and image to the same latent space, the output of both encoders needs to be of the same dimensionality. Hence, it is impossible to combine, for example, a ViT-base for the language part with a ViT-large for the image part.
> We, hence, increased the size of both encoders to ViT-large and report the results below for convenience.
> |          **Model**          | **CLIPMem** | **Lin. Prob. Acc. (ImageNet)** |
> |:---------------------------:|:-----------:|:------------------------------:|
> | Paper (Baseline ViT-Base):  |    0.438    |         63.11% ± 0.91%         |
> |          ViT-Large          |     0.457   |         67.04% ± 1.05%         |
>
> They highlight that while larger encoders improve performance, they also significantly increase memorization. We included this result in the revised version of the paper in Table 5.
>
> >**Most of their findings sound a bit too reasonable and are not surprising. Their finding that augmenting text improves the CLIP model has already been observed in many previous papers though they probably did not discuss memorization.**
>
> We would kindly disagree with the reviewer on the fact that the results are not surprising: For both supervised [1] and self-supervised [2] learning, it has always been shown that decreasing memorization also reduces the performance of the model, i.e., memorization is required for generalization. In contrast, our findings suggest that in CLIP, text augmentation simultaneously reduces memorization while improving performance. This stands in contrast with the other learning paradigms. Additionally, our work is the first to highlight that memorization stems more from the text than the image modality, which provides valuable insights into the inner workings of multi-modal models.
>
> >**Also, it is not hard to imagine that augmenting datasets mitigates memorization.**
>
> By designing a metric to objectively quantify memorization, and by conducting thorough experiments, our work provides scientific evidence for the intuition that augmentations mitigate memorization in multi-modal CLIP models.
>
> >**They mention that their metric is effective in removing noisy samples. But, they compare their approach only with random replacement. I think they need to add more baseline, such as naive CLIP's similarity as done in many works.**
>
> We performed the experiment suggested by the reviewer and also removed samples based on the CLIP similarity score as a baseline to our removal based on CLIPMem. Please see the updated Figure 6. While CLIP similarity also manages to increase performance through removal, it is not as effective as our metric, highlighting the value of considering memorization as a lens to identify noisy samples.
>
> >**In Table 1, the authors augment images by using a diffusion model. But, as they imply, such augmentation can cause a distribution shift in the image side and does not give much intuition about image augmentation.**
>
> To address the reviewer’s concern, we conducted additional experiments. We measured CLIPMem for a model trained with only one generated image and the corresponding real caption. We updated the Table 1 in the paper and here include its copy for the Reviewer’s convenience:
>
> | **Case**                                     | **CLIPMem** | **Lin. Prob. Acc. (ImageNet)** |
> |----------------------------------------------|-------------|--------------------------------|
> |   1 real image + 1 real caption   |    0.438    |         63.11% ± 0.91%         |
> |    1 generated image + 1 real caption      |     0.428   |         63.97% ± 0.79%         |
>
> The results indicate that this augmentation effectively reduces memorization while also providing a slight improvement in performance. Notably, if the distribution shift had been substantial, we would have observed a decrease in performance instead. These findings further reinforce our claims.
>
> **References**
>
> [1] ”Memorization in self-supervised learning improves downstream generalization”. Wenhao Wang, Muhammad Ahmad Kaleem, Adam Dziedzic, Michael Backes, Nicolas Papernot, Franziska Boenisch. ICLR, 2024.
>
> [2] “Does learning require memorization? a short tale about a long tail”. Vitaly Feldman.
> ACM SIGACT Symposium on Theory of Computing, 2020.

---

> > ### Comment · Reviewer_F6JJ · 2024-11-27
> > **after rebuttal**
> >
> > Thanks for the response.
> >
> > I carefully checked the rebuttal, submission, and other reviewers' comments.
> >
> > I understand that the authors made the best effort to address concerns about the weaknesses in experiments.
> >
> > I still have a following concern. In my understanding, memorization should be related to diverse factors, e.g., model size, and noise of data. In this sense, we cannot argue that some techniques, e.g., data augmentation, always help. It is very helpful if we can understand which setting a specific technique is useful in mitigating memorization. However, this work seems to focus on a bit specific setting, e.g., model size. I cannot be sure the generalization of their observation.
> >
> > I would like to keep my rating.

---

> > > ### Author Response · Authors · 2024-11-27
> > > **Thank you**
> > >
> > > We thank the Reviewer for engaging in the discussion with us and checking the rebuttal, submission, and other reviewers’ comments. We are glad that the Reviewer appreciates our work.
> > >
> > > Our work is much broader than the analysis of model size. We **never** *“argue that some techniques, e.g., data augmentation, always help”* but **provide specific parameter values**, e.g., the standard deviation for the added noise, model size (and architecture), and number and strength of augmentations.
> > >
> > > 1. We clearly state that the memorization is related to different factors:
> > > Lines 481-482: “we add small amounts of Gaussian noise to the text embeddings. Our results in Figure 5b and Table 8 highlight that this strategy is highly effective in reducing memorization while improving downstream generalization.” Figure 5 (b) shows that adding the Gaussian noise $N(0,0.15)$ gives us the sweet spot with the smallest memorization and highest performance.
> > > 2. Caption to Figure 5: “(a) We use multiple captions for the same image during training. In our experiments, where we analyzed the range of captions per image from 1 to 5, the case with 5 captions provided the largest reduction of memorization and the biggest increase in performance. (Lines 465-466:) The general trend is that the more captions are used during training, the lower memorization and the higher the linear probing accuracy.
> > > 3. We also observe (in Table 1) that having more augmented images (5 augmented images + 1 caption) helps to decrease memorization while increasing performance (compared to the case with (1 image + 1 caption).
> > > 4. As requested by the Reviewer, we showed that, while keeping the same architecture (ViT), larger encoders (ViT-Large) improve performance and also significantly increase memorization as compared to smaller encoders (ViT-Base).
> > >
> > > Overall, our work considers a broad range of factors and analyzes their impact on memorization. We, therefore, kindly ask the reviewer to re-assess their score.

---

> > > > ### Author Response · Authors · 2024-12-01
> > > > **Have concerns been addressed?**
> > > >
> > > > We would like to follow up on our answers, especially regarding the factors that influence memorization. We demonstrated that (1) adding Gaussian noise (in our experiments: $N(0,0.15)$) to the text embeddings, and (2) increasing the number of captions or images for a given sample, decreases the memorization while enhancing performance. On the other hand, (3) larger encoders (ViT-Large) improve performance and also significantly increase memorization as compared to smaller encoders (ViT-Base). Do our replies adequately address the reviewer's concerns?

---

### Official Review · Reviewer_CAJS · 2024-11-03

**Soundness:** 2
**Presentation:** 3
**Contribution:** 2
**Rating:** 5
**Confidence:** 4

**Summary:**

The paper introduces CLIPMem, a novel metric to quantify memorization in CLIP, which combines elements of supervised and self-supervised learning. The paper shows that memorization within CLIP often arises from mis-captioned or atypical samples, particularly within the text modality rather than the image modality. They propose mitigation strategies that reduce memorization without sacrificing model utility, unlike traditional methods that often degrade performance when reducing memorization. It reports several interesting findings, including 1) CLIP's memorization lies between supervised and self-supervised paradigms, with high memorization for data with inaccurate or misaligned captions; 2) Text domain adjustments, such as using varied or augmented captions, reduce memorization and improve generalization, defying the usual trade-offs seen in other paradigms.

**Strengths:**

+ It introduces a new metric -- CLIPMem to provide a new way for measuring memorization in multi-modal settings, a gap in previous research.
+ It performs empirical analysis to show differences in memorization between the text and image modalities, providing actionable insights.
+ It proposes techniques to successfully reduce memorization while preserving or even enhancing model utility, challenging established norms.
+ By highlighting the risks of training with uncurated, potentially mis-captioned data, the paper suggests guidelines that can benefit real-world multi-modal model training practices.

**Weaknesses:**

- While tailored to CLIP, the metric and findings may need adaptation to apply effectively to other multi-modal models with different architectures.
- The experiments focus on datasets like COCO and CC3M, so it’s unclear how well these findings generalize to other large-scale or domain-specific datasets.
- The mitigation strategies, such as augmenting captions or generating variations, may incur additional computational costs in training, which could limit practicality for some users.

**Questions:**

Would you please comment on how the metric adapt to other multi-modal models besides CLIP?

---

> ### Author Response · Authors · 2024-11-24
> **Adapting the CLIPMem metric & Experiments on the large-scale YFCC100M dataset**
>
> We thank the reviewer for the detailed comments. We are glad that the reviewer recognizes our work as addressing “a gap in previous research” and appreciates that it “provides actionable insights,” “challenges established norms,” and “can benefit real-world multi-modal model training practices.” Below we address all of the points and questions raised by the reviewer one by one.
>
> >**While tailored to CLIP, the metric and findings may need adaptation to apply effectively to other multi-modal models with different architectures. (...) Would you please comment on how the metric adapts to other multi-modal models besides CLIP?**
>
> We tested our metric on the popular CLIP model, but many other multi-modal models follow the CLIP architecture, and our metric is immediately applicable to them as well. For example, multi-modal models with separate encoders and contrastive learning objectives can directly apply CLIPMem with minimal modifications (e.g., ALIGN [1], Florence [2], and LiT [3]), where memorization can be measured by evaluating the alignment scores between representations. For models with additional components other than contrastive alignment, CLIPMem can be applied after alignment before other operations like fusion (ALBEF [4]) or generative tasks (BLIP [5]). By doing so, CLIPMem can isolate and quantify memorization during alignment, being adaptable across different architectures.
>
> >**The experiments focus on datasets like COCO and CC3M, so it’s unclear how well these findings generalize to other large-scale or domain-specific datasets.**
>
> To address the reviewer’s comment, we added additional experiments with the larger YFCC100M dataset. To simulate the large data regime, we trained the model for one epoch, and then evaluate memorization. To ensure comparability to our initial experiments where we trained the model on 70,000 data points for 100 epochs (i.e., 7M samples seen during training), we trained with 7M samples from YFCC100M.
>
> To fit our setup, we trained model f with 6950000 shared +50000 candidate samples and model g with 6950000 shared + 50000 independent samples. We observe that there is still significant memorization when training for one epoch.
>
> |     **Model, Epochs**    | **CLIPMem** | **Lin. Prob. Acc. (ImageNet)** |
> |:------------------------:|:-----------:|:------------------------------:|
> |     YFCC 7M, 1 Epoch     |     0.425   |         64.83% ± 1.04%         |
> | Paper (Coco), 100 Epochs |    0.438    |         63.11% ± 0.91%         |
>
> We also assessed the memorized samples qualitatively, as shown in the new Figure 9 in the updated paper. They highlight that the insights remain the same even when training only for one epoch: atypical and miscaptioned samples are memorized.
>
> **References:**
>
> [1] “Scaling Up Visual and Vision-Language Representation Learning With Noisy Text Supervision” Chao Jia, Yinfei Yang, Ye Xia, Yi-Ting Chen, Zarana Parekh, Hieu Pham, Quoc V. Le, Yunhsuan Sung, Zhen Li, Tom Duerig. ICML 2021.
>
> [2] “Florence: A New Foundation Model for Computer Vision” Lu Yuan, Dongdong Chen, Yi-Ling Chen, Noel Codella, Xiyang Dai, Jianfeng Gao, Houdong Hu, Xuedong Huang, Boxin Li, Chunyuan Li, Ce Liu, Mengchen Liu, Zicheng Liu, Yumao Lu, Yu Shi, Lijuan Wang, Jianfeng Wang, Bin Xiao, Zhen Xiao, Jianwei Yang, Michael Zeng, Luowei Zhou, Pengchuan Zhang.
>  arXiv preprint arXiv:2111.11432 (2021).
>
> [3] “LiT: Zero-Shot Transfer with Locked-image text Tuning” Xiaohua Zhai, Xiao Wang, Basil Mustafa, Andreas Steiner, Daniel Keysers, Alexander Kolesnikov, Lucas Beyer. CVPR 2022.
>
> [4] “Align before Fuse: Vision and Language Representation Learning with Momentum Distillation” Junnan Li, Ramprasaath R. Selvaraju, Akhilesh Deepak Gotmare, Shafiq Joty, Caiming Xiong, Steven Hoi. NeurIPS 2021.
>
> [5] “BLIP: Bootstrapping Language-Image Pre-training for Unified Vision-Language Understanding and Generation” Junnan Li, Dongxu Li, Caiming Xiong, Steven Hoi. ICML 2022.

---

> > ### Author Response · Authors · 2024-11-24
> > **Additional computational costs**
> >
> > >**The mitigation strategies, such as augmenting captions or generating variations, may incur additional computational costs in training, which could limit practicality for some users.**
> >
> > Based on the reviewer’s comment, we benchmarked the generation times:
> > 1. First, we have to point out that augmenting the text in the embedding space takes only 0.016 seconds/caption, i.e., it has a negligible impact on the computational cost or the training time.
> > 2. The generation of additional captions or their paraphrases using LLMs is also very fast (0.28 seconds per paraphrased caption when using GPT3.5). Additionally, this generation incurs zero cost when using open LLMs like Llama, Vicuna, or Mistral.
> >
> > We would like to note that our strategy of adding random noise in the embedding space (see original Table 5b) introduces very small overhead, yet, is effective in limiting memorization.
> >
> > The generation of images is a bit more expensive (on average 1.06 seconds per image when using Stable Diffusion 1.5). However, the generation is done once and then amortizes with more epochs. For the standard CLIP training, a single generated image is reused 35 times. We note that the timing depends on the underlying hardware, especially the GPU used. In our case, we leveraged a single A100 GPU with 80GB of memory. The timing can be further improved using better graphic cards, e.g., the latest H100.

---

> > > ### Author Response · Authors · 2024-12-01
> > > **Have concerns been addressed?**
> > >
> > > We thank the Reviewer for their thoughtful feedback, which has greatly contributed to improving the quality of our paper. We conducted additional experiments and analyses to address all the Reviewer’s concerns and suggestions:
> > >
> > > 1. **CLIPMem is adaptable across different architectures:** Multi-modal models with separate encoders and contrastive learning objectives can directly apply CLIPMem with minimal modifications (e.g., ALIGN [1], Florence [2], and LiT [3]), where memorization can be measured by evaluating the alignment scores between representations. For models with additional components other than contrastive alignment, CLIPMem can be applied after alignment before other operations like fusion (ALBEF [4]) or generative tasks (BLIP [5]).
> > > 2. **Our findings generalize to other datasets:** We train models with infinite data regimes on a 7M subset of YFCC100M, a widely used large-scale multi-modal dataset. Results in Table 6 and Figure 9 show that CLIPMem successfully identifies the most-memorized samples to be miscaptioned samples, confirming that CLIPMem is effective in infinite data regimes and applicable to large-scale, multi-modal datasets beyond COCO and CC3M.
> > > 3. **The proposed methods to successfully reduce memorization while preserving or even enhancing model utility are practical and can benefit real-world multi-modal model training practices:** (1) Augmenting the text in the embedding space takes only 0.016 seconds/caption. (2) The generation of additional captions or their paraphrases using LLMs is also very fast (0.28 seconds per paraphrased caption when using GPT3.5). (3) The generation of images is a bit more expensive (on average 1.06 seconds per image when using Stable Diffusion 1.5), however, the generation is done once and then amortizes with more epochs.
> > >
> > > We hope that our responses address the concerns raised. Therefore, we kindly ask the Reviewer to reconsider their rating in light of these additional insights.
> > >
> > > **References:**
> > >
> > > [1] *“Scaling Up Visual and Vision-Language Representation Learning With Noisy Text Supervision”*. Chao Jia, Yinfei Yang, Ye Xia, Yi-Ting Chen, Zarana Parekh, Hieu Pham, Quoc V. Le, Yunhsuan Sung, Zhen Li, Tom Duerig. ICML 2021.
> > >
> > > [2] *“Florence: A New Foundation Model for Computer Vision”*. Lu Yuan, Dongdong Chen, Yi-Ling Chen, Noel Codella, Xiyang Dai, Jianfeng Gao, Houdong Hu, Xuedong Huang, Boxin Li, Chunyuan Li, Ce Liu, Mengchen Liu, Zicheng Liu, Yumao Lu, Yu Shi, Lijuan Wang, Jianfeng Wang, Bin Xiao, Zhen Xiao, Jianwei Yang, Michael Zeng, Luowei Zhou, Pengchuan Zhang.
> > >  arXiv preprint arXiv:2111.11432 (2021).
> > >
> > > [3] *“LiT: Zero-Shot Transfer with Locked-image text Tuning”*. Xiaohua Zhai, Xiao Wang, Basil Mustafa, Andreas Steiner, Daniel Keysers, Alexander Kolesnikov, Lucas Beyer. CVPR 2022.
> > >
> > > [4] *“Align before Fuse: Vision and Language Representation Learning with Momentum Distillation”*. Junnan Li, Ramprasaath R. Selvaraju, Akhilesh Deepak Gotmare, Shafiq Joty, Caiming Xiong, Steven Hoi. NeurIPS 2021.
> > >
> > > [5] *“BLIP: Bootstrapping Language-Image Pre-training for Unified Vision-Language Understanding and Generation”*. Junnan Li, Dongxu Li, Caiming Xiong, Steven Hoi. ICML 2022.

---

> > > > ### Author Response · Authors · 2024-12-03
> > > > **Feedback on additional analysis and experiments**
> > > >
> > > > We sincerely appreciate the Reviewer's feedback and would like to know if our additional analysis and experiments adequately address the concerns raised.

---

### Official Review · Reviewer_khe4 · 2024-11-09

**Soundness:** 3
**Presentation:** 2
**Contribution:** 3
**Rating:** 6
**Confidence:** 4

**Summary:**

**Summary:** Understanding the memorization / generalization tradeoff is important to properly quantify modern ML models. This work focuses on CLIP which applies InfoNCE between image and text pairs and attempts to quantify the extent to which CLIP models memorize. The authors introduce CLIPMem to quantify memorization and find that the text encoder contributes more to memorization than the vision encoder. They also propose strategies to mitigate / remove memorized samples to improve performance.

**Strengths:**

**Stong points:**
- Well highlighted literature on memorization.
- Defines CLIPMem based on hold one out strategy (similar to Feldman et. al in supervised learning).
- Interesting results on mis-captioned text labels, multi-caption and removal of memorized examples.
- Reasonable pretraining datasets like CC3M
- Clean separation of training and test splits for measuring memorization.

**Weaknesses:**

**Weak points:**
- Missing ability for CLIPMem to be applicable to general off-the-shelf CLIP models. Currently if I understand correctly it requires retraining on specific splits.
- Clarity of specifics of CLIPMem is used for vision only and joint vision + text can be improved.
- The noising results (Table 5-b) are not very convincing. Almost all the results are within the same +/- std range.
- The linear probe accuracy seems quite low (Table 1, 5-a, 5-b,  6-a/b).

**Nit:**
- Text and images in Figure one are very hard to read. Suggest larger and fewer images and move rest to appendix.
- Figure 3 can be make larger / more readable by doing share-y and increasing font sizes.

**Questions:**

**Questions:**

- Is CLIPMem bounded?
- What dimension is kept constant for Table 1? Are the total training samples [count] seen the same?
- It would be interesting to evaluate infinite data regimes (i.e. no repeated data) rather than classical K-epoch  runs. Would the results for - memorization hold here? Just like in language this setting is becoming more and more common.

---

> ### Author Response · Authors · 2024-11-24
> **Applicability of CLIPMem & the noising results**
>
> We thank the reviewer for their insightful comments and are glad that the Reviewer found our results on “mis-captioned text labels, multi-caption and removal of memorization examples” interesting.
>
> >**Missing ability for CLIPMem to be applicable to general off-the-shelf CLIP models. Currently, if I understand correctly it requires retraining on specific splits.**
>
> We would like to clarify that the dependence on retraining on specific splits is an inherent property of the leave-one-out methodology that our metric is based on and not a specific limitation of CLIPMem. This approach is consistent with prior works on measuring memorization in supervised [1] and self-supervised learning [2], which similarly rely on controlled experimental setups in order to accurately test memorization by comparing models trained with and without specific data points.
>
> **References:**
>
> [1] Feldman, Vitaly. "Does learning require memorization? a short tale about a long tail." In Proceedings of the 52nd Annual ACM SIGACT Symposium on Theory of Computing, pp. 954-959. 2020.
>
> [2] Wang, Wenhao, Muhammad Ahmad Kaleem, Adam Dziedzic, Michael Backes, Nicolas Papernot, and Franziska Boenisch. "Memorization in Self-Supervised Learning Improves Downstream Generalization." In The Twelfth International Conference on Learning Representations. 2024
>
> >**Clarity of specifics of CLIPMem is used for vision only and joint vision + text can be improved.**
>
> Existing memorization metrics, such as SSLMem which forms the foundation of our metric, can be used separately for each modality (i.e., either vision or text). However, these metrics are limited when applied to multi-modal models like CLIP, which require interaction between different modalities (as we visualized in Figure 3 in the paper). Hence, our CLIPMem is designed for the joint vision+text multimodal models. Additional clarification would be appreciated if this does not fully address the Reviewer’s concern.
>
> >**The noising results (Table 5-b) are not very convincing. Almost all the results are within the same +/- std range.**
>
> First, we would like to emphasize that the results between no noise, i.e., $\mathcal{N}(0,0)$ and the highest value of noises we evaluated in the original submission $\mathcal{N}(0,0.1)$ differ significantly, indicating a strong (yet continuous) trend through noise addition.
>
> We additionally extended the evaluation and added larger amounts of noise.
>
> | **Noise** | **CLIPMem** | **Lin. Prob. Acc. (ImageNet)** |
> |:---------:|:-----------:|:------------------------------:|
> |    None   |    0.438    |         63.11% ± 0.91%         |
> |  (0,0.01)  |    0.435    |         63.36% ± 0.88%         |
> |  (0,0.05)  |    0.428    |         64.02% ± 1.12%         |
> |  (0,0.10)  |    0.421    |         64.95% ± 0.96%         |
> |  (0,0.15)  |    0.417    |         65.34% ± 0.84%         |
> |  (0,0.20)  |    0.422    |         64.83% ± 0.92%         |
> |  (0,0.25)  |    0.436    |         63.28% ± 0.79%         |
>
> Our results highlight that with $\mathcal{N}(0,0.15)$, the best linear probing accuracy of 65.34% ± 0.84% and the lowest CLIPMem (0.417) can be achieved. This result is significantly outside of the standard deviation of no noise addition with 63.11% ± 0.91%.

---

> > ### Author Response · Authors · 2024-11-24
> > **Linear probe accuracy, Nits, CLIPMem is bounded [-1,+1], dimension in Table 1, Infinite data regimes**
> >
> > >**The linear probe accuracy seems quite low (Table 1, 5-a, 5-b, 6-a/b).**
> >
> > The comparably low linear probing accuracy stems from the significantly smaller scale of our training setup in comparison to the original CLIP model, due to computational constraints. For example, the original CLIP model was trained on approximately 30 million images with a batch size of 1712 [1]. In contrast, our model was trained on only 70000 (65,000 shared and 5,000 candidate) samples with a batch size of 128. As highlighted in prior work, self-supervised learning always needs a large number of training samples [2, 3]. This is the reason for the relatively low linear probing accuracy observed in our experiments. However, since even smaller versions of CLIP demonstrate significant memorization, this effect is likely to be even more pronounced in larger models, as memorization tends to increase with model size [4].
> >
> > We conducted additional experiments where we vary the size of the encoders. Note that since CLIP embeds both text and image to the same latent space, the output of both encoders needs to be of same dimensionality. Hence, it is impossible to combine, for example, a ViT-base for the language part with a ViT-large for the image part.
> >
> > We, hence, increased the size of both encoders to ViT-large and report the results below for convenience.
> >
> > |          **Model**          | **CLIPMem** | **Lin. Prob. Acc. (ImageNet)** |
> > |:---------------------------:|:-----------:|:------------------------------:|
> > | Paper (Baseline ViT-Base):  |    0.438    |         63.11% ± 0.91%         |
> > |          ViT-Large          |     0.457   |         67.04% ± 1.05%         |
> >
> > They highlight that while larger encoders improve performance, they also significantly increase the memorization.
> >
> > **References:**
> >
> > [1] Radford, Alec, et al. "Learning transferable visual models from natural language supervision." International conference on machine learning. PMLR, 2021.
> >
> > [2] Nozawa, Kento, et al. "Understanding Negative Samples in Instance Discriminative Self-supervised Representation Learning." NeurIPS, 2021
> >
> > [3] Liu, Hong, et al. "Self-supervised Learning is More Robust to Dataset Imbalance." ICLR, 2022
> >
> > [4] Wang, Wenhao, et al. “Memorization in Self-Supervised Learning Improves Downstream Generalization.” ICLR, 2024.
> >
> > >**Text and images in Figure one are very hard to read. Suggest larger and fewer images and move rest to appendix. Figure 3 can be make larger / more readable by doing share-y and increasing font sizes.**
> >
> > We updated the paper according to the reviewer’s suggestions and increased the size of images and text (including in Figures 1 and 3).
> >
> > >**Is CLIPMem bounded?**
> >
> > Yes, our CLIPMem is normalized from -1 to 1, and the normalization procedure is described in Appendix A.2, “Normalization on CLIPMem.” A memorization score of $0$ indicates no memorization, +1 indicates the strongest memorization on CLIP model f, and -1 indicates the strongest memorization on CLIP model g.
> >
> > >**What dimension is kept constant for Table 1? Are the total training samples [count] seen the same?**
> >
> > The total number of image-caption pairs is kept the same. We use either 1 or 5 captions from the COCO dataset. For the images, in the case of 1 image, we use the original ones from COCO, and to obtain 5 images per caption, we generate the images for each of the 5 original captions from COCO using Stable Diffusion v1.5.
> >
> > >**It would be interesting to evaluate infinite data regimes (i.e. no repeated data) rather than classical K-epoch runs. Would the results for - memorization hold here? Just like in language this setting is becoming more and more common.**
> >
> > We thank the reviewer for their suggestion and added an additional experiment where we use the larger YFCC100M dataset, trained for one epoch, and then evaluate memorization. To ensure comparability to our initial experiments where we trained the model on 70,000 data points for 100 epochs (i.e., 7M samples seen during training), we trained with 7M samples from YFCC100M.
> >
> > To fit our setup, we trained model f with 6950000 shared + 50000 candidate samples and model g with 6950000 shared + 50000 independent samples. We observe that there is still significant memorization even when training for one epoch.
> >
> > |     **Model** | **Epochs**    | **CLIPMem** | **Lin. Prob. Acc. (ImageNet)** |
> > |:-----------:|:--------:|:-----------:|:------------------------------:|
> > |     YFCC 7M | 1 Epoch     |     0.425   |         64.83% ± 1.04%         |
> > | Paper (Coco) | 100 Epochs |    0.438    |         63.11% ± 0.91%         |
> >
> > We also assessed the memorized samples qualitatively, as shown in the new Figure 9 in the updated paper. They highlight that the insights remain the same even when training only for one epoch: atypical and miscaptioned samples are memorized the most.

---

> > > ### Comment · Reviewer_khe4 · 2024-11-25
> > > **Appreciate the extra ablations and responses**
> > >
> > > We thank the authors for their response and extra experiments (particularly the YFCC 7M run). I'm increaseing my score accordingly as I think memorizing in contrastive vision + language models is quite understudied.

---

> > > > ### Author Response · Authors · 2024-11-25
> > > >
> > > > We appreciate the Reviewer's thoughtful feedback and are glad the additional experiments on YFCC 7M addressed the concerns. We are encouraged by your recognition of the importance of studying memorization in contrastive vision + language models and hope our work contributes to advancing understanding in this area. Thank you for increasing your score and supporting our efforts on this topic.

---

### Author Response · Authors · 2024-11-24
**General Response**

We thank the Reviewers for their thoughtful and encouraging feedback, which greatly helped us further improve our submission. Our work was described as addressing “a gap in previous research” by offering “a new way for measuring memorization in multi-modal settings (Reviewer CAJS). We are encouraged that the significance of the studied trade-off between memorization and generalization was highlighted (Reviewer khe4), with Reviewer CAJS noting our observed trade-off as “challenging established norms” and stating that the paper “suggests guidelines that can benefit real-world multi-modal model training practices.” Reviewer khe4 also found our experimental results *interesting*, especially the removal of memorized mis-captioned examples, which improves performance, underscoring the practical impact of our findings. Finally, Reviewer F6JJ notes that our “insight that memorization is more significant in the text encoder is new” and that our results and analysis “might interest readers.” We hope that our work will significantly contribute to the community and inspire further research, reaching a broader audience.

1. During the rebuttal, we performed additional experiments and analysis that we present in the individual answers to the reviewers, and update them correspondingly in the paper.
2. Based on the suggestion of Reviewer khe4, we experimented with additional noise strength for caption noising and updated the results in Figure 5 (b) and Table 8. When applying random Gaussian noise with 0 mean and 0.15 standard deviation, the linear probing accuracy is 65.34% ± 0.84%, which is out of the +/- std range of baseline’s 63.11% ± 0.91%. This could now prove our statement “Noising the text embedding during training is highly effective in reducing memorization while improving downstream generalization” in the paper.
3. Based on the suggestion of Reviewer F6JJ, we compared the effect of noise-sample removal between CLIPMem with naive CLIP's similarity and updated the results in figures 6 (a) and 6 (b) in the paper. The results show that CLIPMem is more effective than using the naive CLIP’s similarity.
4. Based on the suggestions of Reviewer khe4 and Reviewer CAJS, we train new models with infinite data regimes on a 7M subset of YFCC100M, which is a widely used large-scaled multi-modal dataset. The results in Table 6 and Figure 9 indicate that the most-memorized samples (according to CLIPMem) are still obviously mis-captioned samples, which proves CLIPMem is feasible in infinite data regimes training and can be applied to multi-modal datasets besides COCO and CC3M.
5. Based on the suggestion of Reviewer F6JJ, we performed extra experiments on ViT-large/16 to further study the impact of how model size will affect memorization effect. The results in Table 5 show that with more parameters (larger size), encoders will have higher memorization capacity. This aligns with previous research in Supervised Learning and Self-Supervised Learning [1,2,3].
6. Following Reviewer F6JJ's suggestion, we conducted additional experiments for Table 1 to further substantiate the claim that augmenting either text or images during training significantly reduces memorization.

**References:**

[1] *”Memorization in self-supervised learning improves downstream generalization”.* Wenhao Wang, Muhammad Ahmad Kaleem, Adam Dziedzic, Michael Backes, Nicolas Papernot, Franziska Boenisch. ICLR, 2024.

[2] *“Does learning require memorization? a short tale about a long tail”.* Vitaly Feldman.
ACM SIGACT Symposium on Theory of Computing, 2020.

[3] *“Do ssl models have déjà vu? a case of unintended memorization in self-supervised learning.”* Casey Meehan, Florian Bordes, Pascal Vincent, Kamalika Chaudhuri, Chuan Guo. arXiv e-prints, 2023.

---

### Comment · Area_Chair_Cmp4 · 2024-11-25

Dear Reviewers,

This is a gentle reminder that the authors have submitted their rebuttal, and the discussion period will conclude on November 26th AoE. To ensure a constructive and meaningful discussion, we kindly ask that you review the rebuttal as soon as possible and verify if your questions and comments have been adequately addressed.

We greatly appreciate your time, effort, and thoughtful contributions to this process.

Best regards,
AC

---

### Meta-Review · Area_Chair_Cmp4 · 2024-12-16

**Metareview:**

The authors study memorization on CLIP models and define CLIPMem as a measure of it. They find that "mis-captioned" samples exhibiting highest levels of memorization and that the text encoder contributes more to memorization than the image encoder. They also propose strategies to reduce memorization and improve performance.

This result is interesting since it differs from the behavior of other learning paradigms. Reviewers pointed some areas of improvement like exploring the effect of model size, training on infinite data regimes, or experiments with additional noise strengths. The authors did a good job with the rebuttal, and I believe they succeeded at answering all the reviewer’s concerns. However, while khe4 raised their score, CAJS remained inactive during discussion, and F6JJ decided to keep their score of 5 arguing that results are not surprising.

During reviewer/AC discussion, F6JJ restated their concern and khe4 replied that this work is good science and valuable in itself. I also believe that the fact reducing memorization in CLIP increases performance differs from behavior previously seen in other paradigms.

Overall, and given the enthusiasm of khe4, I believe this is a valuable work and it should be accepted to ICLR 2025.

**Additional Comments On Reviewer Discussion:**

During reviewer/AC discussion, F6JJ restated their concern and khe4 replied that this work is good science and valuable in itself. I also believe that the fact reducing memorization in CLIP increases performance differs from behavior previously seen in other paradigms.

---

### Decision · Program_Chairs · 2025-01-22

Accept (Poster)